# Electromagnetic multipole expansions
# and the logarithmic soft photon theorem

**Geoffrey Compère[*], Dima Fontaine[†] and Kevin Nguyen[‡]**

Université Libre de Bruxelles, BLU-ULB Brussels Laboratory of the Universe,
International Solvay Institutes, CP 231, B-1050 Brussels, Belgium

[*] geoffrey.compere@ulb.be , [†] dima.fontaine@ulb.be , [‡] kevin.nguyen2@ulb.be

## Abstract

We study the general structure of the electromagnetic field in the vicinity of spatial infinity. Starting from the general solution of the sourced Maxwell equations written in terms of multipole moments as obtained by Iyer and Damour, we derive the expansion of the electromagnetic field perturbatively in the electromagnetic coupling. At leading order, where the effect of long-range Coulombic interactions between charged particles is neglected, we discover infinite sets of antipodal matching relations satisfied by the electromagnetic field, which extend and sometimes correct previously known relations. At next-to-leading order, electromagnetic tails resulting from these Coulombic interactions appear, which affect the antipodal matching relations beyond those equivalent to the leading soft photon theorem. Moreover, new antipodal matching relations arise, which we use to re-derive the classical logarithmic soft photon theorem of Sahoo and Sen. Our analysis largely builds upon that of Campiglia and Laddha, although it invalidates the antipodal matching relation which they originally used in their derivation. The antipodal matching relations and the proof of the classical logarithmic soft photon theorem agree with an earlier analysis of Bhatkar, which we generalize using other methods.

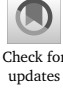

# 1   Introduction and summary of results

It has been established over the last decade that leading soft theorems [1] are consequences of Ward identities associated with asymptotic symmetries [2–6]. The soft theorems are furthermore the Fourier transform in momentum space of position space electromagnetic memory effects [7–9], see [10–12] for reviews. Sub$^n$-leading soft theorems with $n \geq 1$ have also been formulated [13–23] and have been proven to be associated to asymptotic symmetries in a generalized sense [24–36].

The infrared structure of gauge theories strongly differs between tree-level and after including loops. In four spacetime dimensions, the soft expansion of $n$-point amplitudes in QED at tree level admits a Laurent expansion in the soft frequency $\omega$ starting with a pole. Once loop corrections are taken into account, a logarithmic branch $\log \omega$ occurs which multiplies a universal soft factor that only depends upon the momenta, polarization, masses and charges of the incoming and outgoing particles involved [18–21]. This is the logarithmic soft photon theorem. The universal logarithmic soft factor decomposes into two pieces: the quantum part and the classical part. The latter can be obtained from a purely classical computation while the quantum part arises from the computation of Feynman diagrams.

A proposal was made a few years ago in [30] to derive the classical part of the logarithmic soft theorem in QED from the conservation of a charge between future null infinity and past null infinity. However, the authors assumed symmetrically defined asymptotic falloff conditions between future and past null infinity, while an asymmetry occurs due to the retarded nature of the fields. In [23], Bhatkar derived the explicit classical solution to first order in the electromagnetic coupling and derived the corresponding fall-off conditions at future and past null infinities from first principles. In particular, some asymptotic properties of the electromagnetic field can be inferred from the asymptotic expansion of the Liénard-Wiechert solution created by an accelerated charged particle. The resulting electromagnetic field displays an explicit radial logarithmic branch (analogous to the one described in [37] in the gravitational case) not accounted for in the work of [30] and no logarithmic branch in advanced time at past null infinity contrary to that assumed in [30]. He was then able to derive the correct antipodal matching relations [23] which he had shown earlier to imply the logarithmic soft theorem in QED [38]. In this paper, we generalize the work of [30] consistently with the subsequent analysis of Bhatkar [23, 38] to provide an infinite set of antipodal matching relationship up to first order in interactions between matter and the electromagnetic field.

We will make use of the multipolar decomposition of the electromagnetic field introduced by Damour and Iyer [39] as a technique to systematically derive the antipodal matching conditions of the field across spatial infinity. We will proceed perturbatively in the electromagnetic coupling, by first deriving all matching conditions at tree level, before deriving another infi-

nite set of matching conditions at first order in the coupling. One such matching condition will be relevant for the classical logarithmic soft theorem. Our identification of classical charges leading to the logarithmic soft theorem will differ from the one presented in [30] and agree with the ones presented in [23, 38]. Many of the tools and identities developed in [30] will nonetheless play a crucial role in our more general developments. Since our derivation of the antipodal relationships only depends upon the vacuum electromagnetic field equations at spatial infinity, these relationships apply to more generic theories than QED, e.g. charged particles with internal structure, as long as the boundary conditions considered on the induced electromagnetic multipolar moments apply.

The rest of the paper is organized as follows. In Section 2 we derive the asymptotic behavior of the electromagnetic field of a charged particle at future and past null infinities, which is accelerating as a result of its long-range Coulombic interaction with other scattered particles. We use this to derive generic boundary conditions in Section 3, which we express in terms of the multipole moments and their fall-offs in the vicinity of spatial infinity. Under these assumptions we are able to derive the antipodal matching conditions across spatial infinity, first at tree level and second including the effect of interactions among charged particles to first order in the electromagnetic coupling. This will enable us to re-derive the classical logarithmic soft photon theorem in Section 4.

## 2 Asymptotic electromagnetic field of a charged particle

We start by recalling the description of the electromagnetic field given by Damour and Iyer in terms of multipole moments [39], which provide a complete characterization of the solutions to the sourced Maxwell equations. We then turn to the study of the electromagnetic field produced by a charged particle which is still accelerating at asymptotically early/late times as a result of the long-range Coulombic interaction with other charged particles.

### 2.1 Multipole expansion of the electromagnetic field

Following the notations of [39], in four-dimensional Minkowski space described by coordinates $(T, X^i)$ such that the metric is

$$\eta_{\mu\nu}dx^\mu dx^\nu = -dT^2 + (dX^i)^2,\tag{1}$$

the electromagnetic potentials can be expanded in terms of electric and magnetic multipole moments $Q_L(U)$ and $M_L(U)$ as

$$
\begin{aligned}
A_0(T, X^i) &= -\sum_{\ell=0}^{\infty} \frac{(-1)^\ell}{\ell!} \partial_L \left( \frac{Q_L(U)}{R} \right), \\
A_i(T, X^i) &= \sum_{\ell=1}^{\infty} \frac{(-1)^{\ell+1}}{\ell!} \left[ \partial_{L-1} \left( \frac{\overset{(1)}{Q}_{iL-1}(U)}{R} \right) + \frac{\ell}{\ell+1} \varepsilon_{iab} \partial_{aL-1} \left( \frac{M_{bL-1}(U)}{R} \right) \right],
\end{aligned}
\tag{2}
$$

where $U = T - R$, $R^2 = X_i X^i$, and $n_i = X_i/R$ is a unit spatial vector. A repeated capital latin index $L$ implies a $\ell$-dimensional sum over a set of spatial indices $i_1, i_2, ..., i_\ell$, while superscripts denote the number of $U$-derivatives applied on the associated function. Using the identity

$$\partial_L \left( \frac{Q_L}{R} \right) = (-1)^\ell \sum_{j=0}^{\ell} c_{\ell j} \frac{\overset{(\ell-j)}{Q}_L(U)}{R^{j+1}} n_L,\tag{3}$$

given in [40] with $c_{\ell j} \equiv \dfrac{(\ell+j)!}{2^j j! (\ell-j)!}$, the gauge potential can be equivalently written as

$$A_0(T, X^i) = -\sum_{\ell=0}^{\infty} \sum_{j=0}^{\ell} \frac{c_{\ell j}}{\ell!} \frac{\overset{(\ell-j)}{Q_L}(U)}{R^{j+1}} n_L \,, \tag{4}$$

$$A_i(T, X^i) = \sum_{\ell=0}^{\infty} \sum_{j=0}^{\ell} \frac{c_{\ell j}}{(\ell+1)!} \left[ \frac{\overset{(\ell-j+1)}{Q_{iL}}(U)}{R^{j+1}} - \frac{\ell+1}{\ell+2} \varepsilon_{iab} \left( \frac{\overset{(\ell-j+1)}{M_{bL}}(U)}{R^{j+1}} + (j+1+\ell) \frac{\overset{(\ell-j)}{M_{bL}}(U)}{R^{j+2}} \right) n_a \right] n_L \,. \tag{5}$$

We will work in retarded spherical coordinates $(U, R, \theta, \phi)$. We will often denote the angular coordinates by $X^A = (\theta, \phi)$. From $T = U+R$, $X_i = R n_i$ with $n_i = (\sin\theta\cos\phi, \sin\theta\sin\phi, \cos\theta)$, we find

$$F_{UA} = \frac{\partial X^\mu}{\partial U} \frac{\partial X^\nu}{\partial X^A} F_{\mu\nu} = \frac{\partial T}{\partial U} \frac{\partial X^i}{\partial X^A} F_{0i} = R e_A^i F_{0i} \,, \tag{6}$$

$$F_{UR} = \frac{\partial X^\mu}{\partial U} \frac{\partial X^\nu}{\partial R} F_{\mu\nu} = \frac{\partial T}{\partial U} \frac{\partial X^i}{\partial R} F_{0i} = n^i F_{0i} \,, \tag{7}$$

$$F_{RA} = \frac{\partial X^\mu}{\partial R} \frac{\partial X^\nu}{\partial X^A} F_{\mu\nu} = R e_A^i F_{0i} + R n^i e_A^j F_{ij} = F_{UA} + R n^i e_A^j F_{ij} \,, \tag{8}$$

$$F_{AB} = \frac{\partial X^\mu}{\partial X^A} \frac{\partial X^\nu}{\partial X^B} F_{\mu\nu} = \frac{\partial X^i}{\partial X^A} \frac{\partial X^j}{\partial X^B} F_{ij} = R^2 e_A^i e_B^j F_{ij} \,, \tag{9}$$

where we define the field strength as $F_{\mu\nu} = 2\partial_{[\mu} A_{\nu]}$ and $e_A^i \equiv \partial_A n^i$. In electromagnetism, the symplectic flux is determined by the large radius limit of $F_{UA}$. As shown in Appendix A.1, the component $F_{UA}$ is given in terms of the multipole moments by

$$F_{UA} = e_A^i \sum_{\ell=0}^{\infty} \sum_{j=0}^{\ell} \frac{c_{\ell j}}{(\ell+1)!} \left[ \frac{\overset{(\ell-j+2)}{Q_{iL}}(U)}{R^j} - \frac{\ell+1}{\ell+2} \varepsilon_{iab} \left( \frac{\overset{(\ell-j+2)}{M_{bL}}(U)}{R^j} + (j+1+\ell) \frac{\overset{(\ell-j+1)}{M_{bL}}(U)}{R^{j+1}} \right) n_a \right] n_L$$
$$+ \sum_{\ell=0}^{\infty} \sum_{j=0}^{\ell} \frac{c_{\ell j}}{(\ell+1)!} \ell(\ell+1) \frac{\overset{(\ell-j)}{Q_L}(U)}{R^{j+1}} e_{(i_1}^A n_{i_2} ... n_{i_\ell)} \,. \tag{10}$$

An *exactly non-radiative* electromagnetic (EM) field is defined as a field where the energy flux through null infinity vanishes. This amounts to setting the leading $R^0$ term of $F_{UA}$ to zero, see e.g. Eq (3.34) of [41]. Equivalently, an exactly non-radiative EM field has order $\ell$ multipole moments that are polynomials of order $\ell$ at most:

$$Q_L(U) = \sum_{k=0}^{\ell} q_{L,k}^+ U^k \,, \qquad M_L(U) = \sum_{k=0}^{\ell} q_{L,k}^- U^k \,. \tag{11}$$

We will see below that the field sourced by a set of interacting charged particles of charge $e$ is not exactly radiative at spatial infinity, but it is still exactly radiative at linear order in $e$.

## 2.2 Radiation field of interacting charged particles

In this subsection we will study the electromagnetic field produced by a set of charged massive point particles that are accelerating as a result of their Coulombic interactions. More precisely, we will focus on the very late (resp. early) time dynamics where these particles are widely separated and their acceleration is decreasing (resp. increasing) with time as inverse square power. This will give us a benchmark for some of the characteristics expected from a generic

electromagnetic field configuration, as well as prepare the ground for the derivation of the logarithmic soft photon theorem to appear in Section 4.

Thus we consider a set of $n$ charged particles labeled by $a = 1, ..., n$. Each particle is described by a classical trajectory $X_a^\mu(\sigma)$ subject to the equation of motion

$$m_a \frac{d^2 X_a^\mu(\sigma)}{d\sigma^2} = e_a F^\mu{}_\nu(X_a(\sigma)) \frac{dX_a^\nu(\sigma)}{d\sigma}, \tag{12}$$

with $m_a$ the mass and $e_a$ the charge of the particle. Of course the electromagnetic field-strength $F_{\mu\nu}(X)$ is determined through Maxwell equations sourced by the charge current

$$j^\mu(X) = \sum_{a=1}^n e_a \int d\sigma \, \delta^{(4)}(X - X_a(\sigma)) \frac{dX_a^\mu}{d\sigma}. \tag{13}$$

**Asymptotic trajectories**

In the absence of Coulombic interactions the particles would follow straight trajectories. Said differently, the particle's acceleration is proportional to the electric charges. At very late or early proper time ($|\sigma| \to \infty$), the particles are widely separated such that their acceleration is vanishing, and we can effectively use the electric charge as an expansion parameter. Thus, for the trajectories we write

$$X_a^\mu(\sigma) = y_a^\mu + v_a^\mu \sigma + Y_a^\mu(\sigma) + O(e^4), \qquad v_a^2 = -1, \tag{14}$$

where $v_a^\mu$ is the asymptotic velocity and $Y_a^\mu(\sigma)$ is a correction of order $O(e^2)$. To determine the latter, we need to solve (12) using the electromagnetic field sourced by the particles following uncorrected straight trajectories, and given by the Liénard-Wiechert solution

$$F_b^{\mu\nu}(X) = \frac{e_b}{4\pi} \frac{(X - y_b)^\mu v_b^\nu - (X - y_b)^\nu v_b^\mu}{[(v_b \cdot X - v_b \cdot y_b)^2 + (X - y_b)^2]^{3/2}}. \tag{15}$$

As customary, we discard the electromagnetic self-force as it is vanishing faster than the acceleration, such that at order $O(e^2)$ the equation of motion (12) reduces to

$$\begin{aligned} m_a \frac{d^2 Y_a^\mu(\sigma)}{d^2 \sigma} &= \frac{\text{sgn}(\sigma)}{4\pi\sigma^2} \sum_{b \neq a} e_a e_b \frac{(v_a \cdot v_b)(v_a^\mu + y_{ab}^\mu/\sigma) + (1 - v_a \cdot y_{ab}/\sigma)v_b^\mu}{[(v_b \cdot v_a + v_b \cdot y_{ab}/\sigma)^2 + (v_a + y_{ab}/\sigma)^2]^{3/2}} \\ &= \frac{\text{sgn}(\sigma)}{4\pi\sigma^2} \sum_{b \neq a} e_a e_b \frac{(v_a \cdot v_b)v_a^\mu + v_b^\mu}{[(v_a \cdot v_b)^2 - 1]^{3/2}} + O(1/\sigma^3), \end{aligned} \tag{16}$$

where we defined the impact parameters $y_{ab} \equiv y_a - y_b$. The exact solution can be found by integrating the first line of (16), yielding

$$Y_a^\mu(\sigma) = \frac{\text{sgn}(\sigma)}{4\pi} \sum_{b \neq a} e_a e_b \left[ (b_{ab}^\mu - a_{ab}^\mu d_{ab}) t_{ab}(\sigma) + a_{ab}^\mu \log|t_{ab}(\sigma)| \right], \tag{17}$$

where

$$t_{ab}(\sigma) = \frac{d_{ab} + \sigma - \sqrt{e_{ab} + 2d_{ab}\sigma + \sigma^2}}{d_{ab}^2 - e_{ab}}, \tag{18}$$

in terms of the constants

$$a_{ab}^\mu = \frac{(v_a \cdot v_b)v_a^\mu + v_b^\mu}{[(v_b \cdot v_a)^2 - 1]^{3/2}}, \qquad b_{ab}^\mu = \frac{(v_a \cdot v_b)y_{ab}^\mu - (v_a \cdot y_{ab})v_b^\mu}{[(v_b \cdot v_a)^2 - 1]^{3/2}}, \qquad (19)$$

$$d_{ab} = \frac{(v_a \cdot v_b)(v_b \cdot y_{ab}) + v_a \cdot y_{ab}}{[(v_b \cdot v_a)^2 - 1]^{3/2}}, \qquad e_{ab} = \frac{(v_b \cdot y_{ab})^2 + y_{ab}^2}{[(v_b \cdot v_a)^2 - 1]^{3/2}}. \qquad (20)$$

Note that $Y_a \cdot v_a = 0$. The asymptotic solution in the limit $|\sigma| \to \infty$ is thus found to be

$$Y_a^\mu(\sigma) = c_a^\mu \operatorname{sgn}(\sigma) \ln|\sigma| + O(1/\sigma), \qquad (21)$$

with

$$c_a^\mu = -\frac{1}{4\pi m_a} \sum_{b \neq a} e_a e_b \frac{(v_b \cdot v_a)v_a^\mu + v_b^\mu}{[(v_b \cdot v_a)^2 - 1]^{3/2}}. \qquad (22)$$

This leading logarithmic correction can already be found in [18, 42]. The corrected velocity and acceleration to order $O(e^2)$ are therefore given by

$$V_a^\mu(\sigma) = v_a^\mu + c_a^\mu |\sigma|^{-1} + O(1/\sigma^2), \qquad A_a^\mu(\sigma) = -c_a^\mu \operatorname{sgn}(\sigma)|\sigma|^{-2} + O(1/\sigma^3), \qquad (23)$$

and the acceleration indeed vanishes as $|\sigma|^{-2}$ as anticipated. We note that to this order the velocity is normalized ($V_a^2 = -1$) thanks to the property $c_a \cdot v_a = 0$.

**Liénard-Wiechert field of accelerated particles**

Having determined the leading correction to the particle trajectories, we can then evaluate the leading correction to the radiated electromagnetic field. To do so we use the general Liénard-Wiechert solution sourced by a single charged point particle following an arbitrary trajectory $X^\mu(\sigma)$, namely [43]

$$F^{\mu\nu}(X) = -\frac{e}{4\pi} \frac{1}{V(\sigma) \cdot (X - X(\sigma))} \frac{d}{d\sigma} \left[ \frac{(X - X(\sigma))^\mu V^\nu(\sigma) - (\mu \leftrightarrow \nu)}{V(\sigma) \cdot (X - X(\sigma))} \right]_{\sigma = \sigma_*(X)}, \qquad (24)$$

with $\sigma_*(X)$ the proper time value such that $X(\sigma^*)$ lies on the past lightcone of the evaluation point $X$,

$$[X - X(\sigma_*)]^2 = 0, \qquad T > T(\sigma_*). \qquad (25)$$

Again we solve this equation perturbatively in $e^2$, using the asymptotic trajectory (14) and dropping the index $a$ for convenience. It is relatively straightforward to show that the leading and subleading contributions in $e^2$ are given by

$$\sigma_*(X) = \bar{\sigma}_*(X) + \delta\sigma_*(X) + O(e^4), \qquad (26)$$

with

$$\bar{\sigma}_* = v \cdot y - v \cdot X - \sqrt{(v \cdot X - v \cdot y)^2 + (X - y)^2}, \qquad \delta\sigma_* = -\frac{(X - y) \cdot Y(\bar{\sigma}_*)}{\bar{\sigma}_* + (X - y) \cdot v}. \qquad (27)$$

For an incoming particle ($\sigma \to -\infty$), this allows us to evaluate the radiated electromagnetic field in a neighborhood of past null infinity $\mathscr{I}^-$. To achieve this we switch to advanced coordinates $T = V - R, X^i = R n^i$ and consider the large $R$ limit for which we find

$$\bar{\sigma}_* = -2R(v^0 + n_i v^i) + O(R^0), \qquad \delta\sigma_* = \ln R \frac{c^0 + n_i c^i}{v^0 + n_i v^i} + O(R^0). \qquad (28)$$

For convenience we can introduce the future-directed null vector $n^\mu = (1, -n_i)$ such that these can be written in covariant form

$$\bar{\sigma}_* = 2R\,n\cdot v + O(R^0), \qquad \delta\sigma_* = \ln R\,\frac{n\cdot c}{n\cdot v} + O(R^0). \tag{29}$$

Since $n\cdot v < 0$, we can confirm that $\sigma_*(X) \to -\infty$ as $R \to \infty$, which validates the use of the asymptotic solution described above in this limit. Using this one can evaluate (24) near $\mathscr{I}^-$. It is interesting to notice that the acceleration $A^\mu(\sigma)$ does not contribute to leading order and first logarithmic correction $R^{-1}\ln R$, such that to these orders we can use the simpler expression

$$F^{\mu\nu}(X) = \frac{e}{4\pi}\,\frac{(X - X(\sigma_*))^\mu V^\nu(\sigma_*) - (X - X(\sigma_*))^\nu V^\mu(\sigma_*)}{[V(\sigma_*)\cdot(X - X(\sigma_*))]^3}. \tag{30}$$

Let us look in particular at the component $F_{RA}$ as it will play an important role in the derivation of the logarithmic soft photon theorem. For the denominator we have

$$\begin{aligned}
V(\sigma_*)\cdot(X - X(\sigma_*)) &= v\cdot(X - y) + \sigma_* + c\cdot X\,\sigma_*^{-1} + O(e^4) \\
&= v\cdot(X - y) + \bar{\sigma}_* + \delta\sigma_* + c\cdot X\,\bar{\sigma}_*^{-1} + O(e^4) \\
&= R\,n\cdot v + \ln R\,\frac{n\cdot c}{n\cdot v} + O(R^0) + O(e^4),
\end{aligned} \tag{31}$$

such that we find

$$F_{RA} = R(-e_A^i F_{0i} + n^i e_A^j F_{ij}) = R^{-2}\ln R\,\frac{e}{4\pi}\,\frac{e_A^\mu\,(v_\mu c_\nu - v_\nu c_\mu)\,n^\nu}{(n\cdot v)^3} + O(R^{-2}), \tag{32}$$

where we have introduced $e_A^\mu = (0, e_A^i)$ for covariance. Rather interestingly we witness the appearance of $\ln R$ terms in the expansion of the field-strength near $\mathscr{I}^-$. This is the electromagnetic analogue of the 'loss of peeling' discussed by Damour in the gravitational context [37].

**Radiative multipoles**

We can also investigate the multipolar structure of the electromagnetic field near future null infinity $\mathscr{I}^+$ directly, using their relation to the charge current [39],

$$Q_L(U) = N_\ell \int d^3x\,\hat{x}_L \int_{-1}^1 dz\,(1 - z^2)^\ell \left[(\ell + 1)\tilde{j}^0 + R(z\partial_U \tilde{j}^0 - \partial_U \tilde{j}^i n_i)\right], \tag{33}$$

with $N_\ell \equiv (2\ell + 1)!!/2^{\ell+1}(\ell + 1)!$ and $\tilde{j} \equiv j(\vec{X}, U + Rz)$ the retarded current. The hatted notation stands for the symmetric tracefree part (STF), i.e., $\hat{x}_L = x_{\langle i_1 \dots i_\ell \rangle}$. Using now $\partial_U \tilde{j}^\mu = R^{-1}\partial_z \tilde{j}^\mu$ and integrating by parts, we can also write (33) as

$$Q_L(U) = \delta_{\ell,0}Q_0(U) + N_\ell\,\ell \int d^3x\,\hat{x}_L \int_{-1}^1 dz\,(1 - z^2)^{\ell-1}\left[(1 + z^2)\tilde{j}^0 - 2z\,\tilde{j}^i n_i\right], \tag{34}$$

where the total electric charge can be computed as

$$\begin{aligned}
Q_0(U) &= \frac{1}{2}\int d^3x\left[z\tilde{j}^0 - \tilde{j}^i n_i\right]_{-1}^1 = \frac{1}{2}\int d^3x\left[J^*(\vec{X}, U + R) + J^*(\vec{X}, U - R)\right] \\
&= \int d^3x\,J^*(\vec{X}, U + Rz).
\end{aligned} \tag{35}$$

Here we introduced the charge density

$$J^*(\vec{X}, U + Rz) \equiv \tilde{j}^0 - z\tilde{j}^i n_i, \tag{36}$$

whose integral over space is independent of $z$ on account of the current conservation written in the form [39]

$$\partial_z J^* = -\frac{d}{dx^i}(R\tilde{j}^i).\tag{37}$$

The current corresponding to a point particle is given by (13), such that

$$\tilde{j}^\mu = j^\mu(\vec{X}, U + Rz) = e\int d\sigma\,\frac{\delta^{(3)}(\vec{X} - \vec{X}(\sigma))}{R}\,\delta\left(z - \frac{X^0(\sigma) - U}{r}\right)\frac{dX^\mu}{d\sigma},\tag{38}$$

and the multipole moments therefore read, for $\ell \neq 0$,

$$Q_L(U) = N_\ell\,\ell\,e\int d\sigma\,\frac{\hat{X}_L(\sigma)}{R(\sigma)}(1-z^2)^{\ell-1}\left[(1+z^2)\frac{dX^0}{d\sigma} - 2z\frac{dX^i}{d\sigma}\frac{X_i(\sigma)}{R(\sigma)}\right],\tag{39}$$

with

$$z(\sigma) = \frac{X^0(\sigma) - U}{R(\sigma)}.\tag{40}$$

Up to this point we have made no approximation. Let us now evaluate the multipole moments produced by a point particle following the accelerated trajectory (14). To simplify the computation we can use a translation in order to set to zero the origin of the reference trajectory $y_a^\mu = 0$ (note that the distances $y_{ab}^\mu$ are invariant however). Up to order $e^3$ (remember $Y^\mu = O(e^2)$) we have

$$\begin{aligned}Q_L(U) &= N_\ell\,\ell\,e\,\frac{\hat{v}_L}{|\vec{v}|}\int d\sigma\,\sigma^{\ell-1}(1-z^2)^{\ell-1}\left[(1+z^2)v^0 - 2z\,|\vec{v}|\right]\\[4pt]&\quad + N_\ell\,\ell\,e\,\frac{\hat{v}_L}{|\vec{v}|}\int d\sigma\,\sigma^{\ell-1}(1-z^2)^{\ell-1}\left[(1+z^2)\dot{Y}^0(\sigma) - 2z\,\frac{\dot{Y}^i(\sigma)v_i}{|\vec{v}|}\right]\\[4pt]&\quad + N_\ell\,\ell^2\,e\int d\sigma\,\frac{v_{\langle L-1}Y_{L\rangle}(\sigma)}{|\vec{v}|}\sigma^{\ell-2}(1-z^2)^{\ell-1}\left[(1+z^2)v^0 - 2z\,|\vec{v}|\right]\\[4pt]&\quad - N_\ell\,\ell\,e\,\frac{\hat{v}_L}{|\vec{v}|^3}\int d\sigma\,Y^i(\sigma)v_i\,\sigma^{\ell-2}(1-z^2)^{\ell-1}\left[(1+z^2)v^0 - 2z\,|\vec{v}|\right],\end{aligned}\tag{41}$$

with $z$ expanded as

$$z(\sigma) = \frac{v^0\sigma - U}{|\vec{v}|\sigma} + \delta z(\sigma) + O(e^4),\qquad \delta z(\sigma) \equiv \frac{Y^0(\sigma)|\vec{v}|^2\sigma - (v^0\sigma - U)v_iY^i(\sigma)}{|\vec{v}|^3\sigma^2}.\tag{42}$$

We can invert this relation,

$$\sigma(z) = \bar{\sigma}(z) + \delta\sigma(z) + O(e^4),\tag{43}$$

with

$$\begin{aligned}\bar{\sigma}(z) &= \frac{U}{v^0 - |\vec{v}|z},\\[4pt]\delta\sigma(z) &= -\frac{|\vec{v}|\bar{\sigma}(z)^2}{U}\delta z(\bar{\sigma}(z)) = \left.\frac{(v^0\sigma - U)v_iY^i(\sigma) - Y^0(\sigma)|\vec{v}|^2\sigma}{|\vec{v}|^2 U}\right|_{\sigma = \bar{\sigma}(z)}.\end{aligned}\tag{44}$$

Using this to change the integration variable in (41), to order $O(e^3)$ we get

$$
\begin{aligned}
Q_L(U) = &\; N_\ell \, \ell \, e \, \hat{v}_L \, U^\ell \int_{-1}^{1} dz \, (v^0 - |\vec{v}|z)^{-(\ell+1)} (1-z^2)^{\ell-1} \left[ (1+z^2)v^0 - 2z|\vec{v}| \right] \\
&+ N_\ell \, \ell \, e \, \hat{v}_L \, U^\ell \int_{-1}^{1} dz \, \frac{d\delta\sigma}{d\bar{\sigma}} (v^0 - |\vec{v}|z)^{-(\ell+1)} (1-z^2)^{\ell-1} \left[ (1+z^2)v^0 - 2z|\vec{v}| \right] \\
&+ N_\ell \, \ell(\ell-1) \, e \, \hat{v}_L \, U^{\ell-1} \int_{-1}^{1} dz \, \delta\sigma \, (v^0 - |\vec{v}|z)^{-\ell} (1-z^2)^{\ell-1} \left[ (1+z^2)v^0 - 2z|\vec{v}| \right] \\
&+ N_\ell \, \ell \, e \, \hat{v}_L \, U^\ell \int_{-1}^{1} dz \, (v^0 - |\vec{v}|z)^{-(\ell+1)} (1-z^2)^{\ell-1} \left[ (1+z^2)\dot{Y}^0(\bar{\sigma}) - 2z \frac{\dot{Y}^i(\bar{\sigma})v_i}{|\vec{v}|} \right] \\
&+ N_\ell \, \ell^2 \, e \, U^{\ell-1} \int_{-1}^{1} dz \, v_{\langle L-1} Y_{L\rangle}(\bar{\sigma}) (v^0 - |\vec{v}|z)^{-\ell} (1-z^2)^{\ell-1} \left[ (1+z^2)v^0 - 2z|\vec{v}| \right] \\
&- N_\ell \, \ell \, e \, \frac{\hat{v}_L}{|\vec{v}|^2} U^{\ell-1} \int_{-1}^{1} dz \, Y^i(\bar{\sigma}) v_i \, (v^0 - |\vec{v}|z)^{-\ell} (1-z^2)^{\ell-1} \left[ (1+z^2)v^0 - 2z|\vec{v}| \right],
\end{aligned}
\tag{45}
$$

with $\bar{\sigma}(z)$ and $\delta\sigma(z)$ given in (44), and

$$
\frac{d\delta\sigma}{d\bar{\sigma}} = \frac{(v^0\bar{\sigma} - U)v_i \dot{Y}^i(\bar{\sigma}) + v^0 v_i Y^i(\bar{\sigma}) - |\vec{v}|^2 (\bar{\sigma}\,\dot{Y}^0(\bar{\sigma}) + Y^0(\bar{\sigma}))}{|\vec{v}|^2 U}.
\tag{46}
$$

This is a fairly complicated expression, depending on the exact solution $Y_a^\mu(\sigma)$ given in (17). Of interest to us is the structure of the multipoles as a function of the retarded time $U$, which takes the form

$$
Q_L(U) = \sum_{k=0}^{\ell} q_{L,k}^{(0)} U^k + \sum_{k=1}^{\infty} q_{L,k}^{(\ln)} U^{\ell-k} \ln U + \sum_{k=1}^{\infty} q_{L,k}^{(1)} U^{\ell-k}.
\tag{47}
$$

Here the factors $q_{L,k}^{(0)}$ are of the order of the electromagnetic coupling $e$ while the first radiative corrections $q_{L,k}^{(\ln)}$ and $q_{L,k}^{(1)}$ are of order $e^3$. To be fully general we should also reinstate the dependence on the origin of the trajectory $y_a^\mu$, which we can achieved by performing the coordinate transformation

$$
U \mapsto U + y_a^0 + y_a^i n_i + O(R^{-1}).
\tag{48}
$$

This will not affect the coefficients of the expansion (47) that dominate the regime $|U| \to \infty$, and which we can give explicitly fo future reference, namely

$$
q_{L,\ell}^{(0)} = N_\ell \, \ell \, e \, \hat{v}_L \int_{-1}^{1} dz \, (v^0 - |\vec{v}|z)^{-(\ell+1)} (1-z^2)^{\ell-1} \left[ (1+z^2)v^0 - 2z|\vec{v}| \right],
\tag{49}
$$

and

$$
\begin{aligned}
q_{L,1}^{(\ln)} = &\; N_\ell \, \ell^2 \, e \, \hat{v}_L \, \frac{\tilde{v} \cdot c}{|\vec{v}|^2} \operatorname{sgn}(U) \int_{-1}^{1} dz \, (v^0 - |\vec{v}|z)^{-(\ell+1)} (1-z^2)^{\ell-1} \left[ (1+z^2)v^0 - 2z|\vec{v}| \right] \\
&- N_\ell \, \ell^2 \, e \, \hat{v}_L \, \frac{v^i c_i}{|\vec{v}|^2} \operatorname{sgn}(U) \int_{-1}^{1} dz \, (v^0 - |\vec{v}|z)^{-\ell} (1-z^2)^{\ell-1} \left[ (1+z^2)v^0 - 2z|\vec{v}| \right] \\
&+ N_\ell \, \ell^2 \, e \, v_{\langle L-1} c_{L\rangle} \operatorname{sgn}(U) \int_{-1}^{1} dz \, (v^0 - |\vec{v}|z)^{-\ell} (1-z^2)^{\ell-1} \left[ (1+z^2)v^0 - 2z|\vec{v}| \right],
\end{aligned}
\tag{50}
$$

where we have introduced the unit spacelike vector $\tilde{v}^\mu = (|\vec{v}|^2, v^0 \vec{v})$ for convenience. These coefficients are only sensitive to the velocities $v_a$ of the charged particles but not of their impact parameters $y_{ab}$.

# 3 Antipodal matchings from multipole expansions

Due to the linearity (in the multipole moments) of the equations at hand, our analysis can be broken down in a non-radiative case with terms of order $O(e)$, and in radiative corrections of order $O(e^3)$ dominated by a logarithmic term at large $U$. Thus we consider the multipole moments

$$Q_L(U) = \sum_{k=0}^{\ell} q_{L,k}^{(0)} U^k + q_{L,1}^{(\ln)} \ln|U| U^{\ell-1} + O(e^3, U^{\ell-1}), \tag{51}$$

where $q_{L,k}^{(0)} \sim O(e)$ and $q_{L,1}^{(\ln)} \sim O(e^3)$. As is apparent from Eq. (2), the field strength will involve derivatives of the multipole moments. These are given by

$$\overset{(p)}{Q}_L(U) = \sum_{k=0}^{\ell} q_{L,k}^{(0)} \frac{k!}{(k-p)!} U^{k-p} \Theta_{k-p} + q_{L,1}^{(\ln)} \frac{(\ell-1)!}{(\ell-1-p)!} \ln(|U|) U^{\ell-1-p} \Theta_{\ell-1-p} + O(U^{\ell-1-p}), \tag{52}$$

where $\Theta_n$ is the discrete Heaviside distribution defined by

$$\Theta_n = \begin{cases} 1, & \text{if } n \geq 0, \\ 0, & \text{otherwise.} \end{cases} \tag{53}$$

## 3.1 Non-radiative case: Antipodal matchings across spatial infinity

In this subsection, we explore matching relations across spatial infinity, for all components of the field strength, stemming *only* from the tree-level $O(e)$ contribution of Eq. (51).

**Radial electric field $F_{UR}$.** The radial component of the electric field is captured in the $F_{UR}$ component of the field strength. As shown in Eq. (A.8) of Appendix A.1, in a non-radiative setting this component can be written

$$F_{UR} = \sum_{n=0}^{\infty} \frac{1}{R^{n+2}} \overset{n}{F}_{UR} = \sum_{n=0}^{\infty} \sum_{k=0}^{n} \frac{1}{R^{n+2}} U^{n-k} \overset{n,k}{F}_{UR}. \tag{54}$$

Expanded in terms of multipoles, we have

$$\overset{n}{F}_{UR} = \sum_{\ell=n}^{\infty} \frac{c_{\ell+1,n+1}}{(\ell+2)!} \left( \overset{(\ell-n+2)}{Q}_{L+2}(U) n_{L+2} - (\ell+2) \overset{(\ell-n+1)}{Q}_{L+1}(U) n_{L+1} \right) - \frac{c_{\ell n}}{\ell!} (n+1) \overset{(\ell-n)}{Q}_L(U) n_L. \tag{55}$$

It is convenient to project the field onto spherical harmonics. Projected on a given harmonics $n_{L'}$, with $\langle n_L | n_{L'} \rangle = \mathcal{C}_\ell \delta_{\ell\ell'}$, we find

$$\langle \overset{n}{F}_{UR}, n_L \rangle = \frac{\mathcal{C}_\ell}{\ell!} \overset{(\ell-n)}{Q}_L(U) \left( c_{\ell-1,n+1} - c_{\ell,n+1} - (n+1) c_{\ell n} \right). \tag{56}$$

As shown by Eq. (11), the non-radiative condition at $i^0$ constrains the multipole moments to be polynomials. Therefore, the tree-level contribution to $F_{UR}$ is given by

$$\langle \overset{n,k}{F}_{UR}, n_L \rangle = \frac{\mathcal{C}_\ell}{\ell!} \frac{(\ell-k)!}{(n-k)!} q_{L,\ell-k}^{+,(0)} \left( c_{\ell-1,n+1} - c_{\ell,n+1} - (n+1) c_{\ell,n} \right). \tag{57}$$

As mentioned in Appendix A.1, upon the change of coordinates $U = V - 2R$, the field strength becomes

$$F_{VR} = \sum_{n=0}^{\infty} \sum_{k=0}^{n} \frac{1}{R^{n+2}} V^{n-k} \overset{n,k}{F}_{VR}, \tag{58}$$

with

$$\langle \overset{n,k}{F_{VR}}, n_L \rangle = \sum_{j=0}^{\infty} (-2)^j \binom{n+j-k}{j} \langle \overset{n+j,k}{F_{UR}}, n_L \rangle. \tag{59}$$

By explicit computation of the respective projections of $\overset{n,k}{F_{VR}}(n^i)$ and $\overset{n,k}{F_{UR}}(-n^i)$ on every level-$\ell$ harmonic, we find the important relation

$$\overset{n,k}{F_{VR}}(n^i)\Big|_{\mathscr{I}_+^-} = (-1)^n \overset{n,k}{F_{UR}}(-n^i)\Big|_{\mathscr{I}_-^+}, \qquad \forall\, n \geq 0, \quad 0 \leq k \leq n, \tag{60}$$

where $n^i$ and $-n^i$ are antipodal points on the sphere. Hence, we find an infinite set of antipodal matching relations for the radial electric field, to all orders in $R$ and $U$. We note that for $n = k$, this result was presented in [29] up to the factor $(-1)^n$. We investigated the origin of this sign discrepancy and corrected the reasoning of [29], which we detail in Appendix B.

**Radial magnetic field $F_{AB}$.** The existence of antipodal matching conditions to any order in $R$ and $U$ for the radial electric field suggests that similar matching conditions also exist for the magnetic field, in the non-radiative case. In Appendix A.2, we show that at first order in the coupling constant, $F_{AB}$ can be expanded as

$$F_{AB} = \sum_{n=0}^{\infty} \frac{1}{R^n} \overset{n}{F_{AB}} = \sum_{n=0}^{\infty} \sum_{k=0}^{n} \frac{1}{R^n} U^{n-k} \overset{n,k}{F_{AB}}\Big|_{\mathscr{I}_-^+}. \tag{61}$$

As $F_{AB}$ is antisymmetric, all projections onto symmetric tensor spherical harmonics will trivially vanish. Hence, the only non-trivial projection will involve the Levi-Civita symbol $\epsilon^{AB}$. Using the standard choice of the relative orientation of the sphere with respect to the orientation of Euclidean space,

$$\varepsilon^{AB} e_A^i e_B^j = \epsilon_{ijk} n^k, \tag{62}$$

we find

$$\varepsilon^{AB} \overset{n}{F_{AB}} = \sum_{\ell=0}^{\infty} \frac{1}{(\ell+n+1)!} c_{\ell+n,n} (\ell+n+1) n_{L+N+1} \overset{(\ell+1)}{M_{L+N+1}}(U)$$
$$+ \sum_{\ell=0}^{\infty} \frac{1}{(\ell+n)!} c_{\ell+n-1,n-1} (\ell+n)(2n+\ell-1) n_{L+N} \overset{(\ell)}{M_{L+N}}(U). \tag{63}$$

Projected on arbitrary harmonics $n_L$, we find

$$\langle \epsilon^{AB} \overset{n}{F_{AB}}, n_L \rangle = \frac{\mathcal{C}_\ell}{(\ell-1)!} (c_{\ell-1,n} + (n+\ell+1)c_{\ell-1,n-1}) \overset{(\ell-n)}{M_L}(U). \tag{64}$$

Plugging in the non-radiative constraint (11) then results in

$$\langle \varepsilon^{AB} \overset{n,k}{F_{AB}}\Big|_{\mathscr{I}_-^+}, n_L \rangle = \frac{\mathcal{C}_\ell}{(\ell-1)!} \frac{(\ell-k)!}{(n-k)!} q_{L,\ell-k}^{-,(0)} (c_{\ell-1,n} + (n+\ell+1)c_{\ell-1,n-1}). \tag{65}$$

Similarly to the radial electric field, using the transformation $U \to V - 2R$, we find

$$F_{AB} = \sum_{n=0}^{\infty} \sum_{k=0}^{n} \frac{1}{R^n} V^{n-k} \overset{n,k}{F_{AB}}\Big|_{\mathscr{I}_+^-}, \tag{66}$$

with

$$\overset{n,k}{F_{AB}}\Big|_{\mathscr{I}_+^-} = \sum_{j=0}^{\infty} (-2)^j \binom{n+j-k}{j} \overset{n,k}{F_{AB}}\Big|_{\mathscr{I}_-^+}. \tag{67}$$

Just as for the radial electric field, we find the following infinite set of antipodal matching relations across spatial infinity,

$$\overset{n,k}{F_{AB}}(n^i)\Big|_{\mathscr{I}_+^-} = (-1)^n \overset{n,k}{F_{AB}}(-n^i)\Big|_{\mathscr{I}_-^+}, \qquad \forall\, n \geq 0, \quad 0 \leq k \leq n. \tag{68}$$

**Transverse electric field $F_{UA}$.** We can also show that antipodal matching relations across $i^0$ exist for the remaining components of the field strength. Let us start with the transverse electric field $F_{UA}$. We define the expansions

$$F_{UA} = \sum_{n=0}^{\infty}\sum_{k=0}^{n}\frac{1}{R^{n+1}}U^{n-k}\overset{n+1,k+1}{F}_{UA} = \sum_{n=0}^{\infty}\sum_{k=0}^{n}\frac{1}{R^{n+1}}V^{n-k}\overset{n+1,k+1}{F}_{VA} = F_{VA}. \tag{69}$$

The exact expansion for $F_{UA}$ is given by Eq. (A.5). Let us first consider the electric contribution by setting the magnetic multipole moments to zero. We will restore them at the end. Then, we find

$$F_{UA} = \sum_{j=0}^{\infty}\sum_{\ell=j}^{\infty}\frac{c_{\ell j}}{(\ell+1)!}\left[\frac{\overset{(\ell-j+2)}{Q_{iL}}(U)}{R^j}e_A^i n_L + \ell(\ell+1)\frac{\overset{(\ell-j)}{Q_L}(U)}{R^{j+1}}e_{(i_1}^A n_{i_2}...n_{i_\ell)}\right]$$

$$= \sum_{j=0}^{\infty}\sum_{\ell=j}^{\infty}\frac{1}{(\ell+1)!}\left[c_{\ell,j+1}\frac{\overset{(\ell-j+1)}{Q_{iL}}(U)}{R^{j+1}}e_A^i n_L + c_{\ell j}(\ell+1)\frac{\overset{(\ell-j)}{Q_L}(U)}{R^{j+1}}\partial_A n_L\right], \tag{70}$$

that is

$$\overset{n+1}{F}_{UA} = \sum_{\ell=n}^{\infty}\frac{1}{(\ell+1)!}\left[c_{\ell,n+1}\overset{(\ell-n+1)}{Q_{iL}}(U)e_A^i n_L + c_{\ell n}(\ell+1)\overset{(\ell-n)}{Q_L}(U)\partial_A n_L\right]. \tag{71}$$

This has to be projected on even or odd vector harmonics $D^A n_{L'}$ or $\epsilon^{AB}D_B n_{L'}$, respectively. We can show that the projection onto the odd vector harmonics vanishes as a result of the antisymmetry of the Levi-Civita tensor and the symmetry of the multipoles. Thus, only the even projection remains, which, upon integration by parts, boils down to computing the divergence of (71). With $D^A D_A n_L = -\ell(\ell+1)n_L$, this is

$$D^A\overset{n+1}{F}_{UA} = \sum_{\ell=n}^{\infty}\left[\frac{c_{\ell,n+1}}{(\ell+1)!}\overset{(\ell-n+1)}{Q_{iL}}(U)(-2n_i n_L + D_A n^i D^A n_L) - \frac{c_{\ell,n}}{\ell!}\ell(\ell+1)\overset{(\ell-n)}{Q_L}(U)n_L\right]. \tag{72}$$

Since $Q_{iL}(U)$ is totally symmetric, we have

$$Q_{iL}(U)\int d\Omega\, D_A n_i\, D^A n_L\, n_{L'} = \ell\, Q_{ijL-1}(U)\int d\Omega\, D_A n_i\, D^A n_j\, n_{L-1}\, n_{L'}$$

$$= -\ell\, Q_{L+1}(U)\int d\Omega\, n_{L+1}\, n_{L'}, \tag{73}$$

where we used the identity $D_A n_i D^A n_j = \delta_{ij} - n_i n_j$ and the tracelessness of the multipole moments. Then,

$$\langle\overset{n+1}{F}_{UA}, D^A n_L\rangle = \frac{C_\ell}{\ell!}\overset{(\ell-n)}{Q_L}(U)\left(c_{\ell-1,n+1}(\ell+1) + c_{\ell,n}\ell(\ell+1)\right). \tag{74}$$

Plugging in the tree-level multipole moment, exactly as in the two previous cases, one finds the following matching relations across spatial infinity,

$$\overset{n+1,k+1}{F}_{UA}(n^i)\Big|_{\mathscr{I}_-^+} = (-1)^n \overset{n+1,k+1}{F}_{VA}(-n^i)\Big|_{\mathscr{I}_+^-}, \qquad \forall\, n\geq 0, \quad 0\leq k\leq n, \tag{75}$$

where the coefficients of the expansions have been found to satisfy

$$\overset{n+1,k+1}{F}_{VA}\Big|_{\mathscr{I}_+^-} = \sum_{j=0}^{\infty}(-2)^j\binom{n+j-k}{j}\overset{n+j+1,k+1}{F}_{UA}\Big|_{\mathscr{I}_-^+}. \tag{76}$$

By explicit computation, we were able to derive the result (75) in the presence of all magnetic multipole moments as well. The derivation is similar to the one explicitly displayed here.

**Component $F_{RA}$.** Finally we turn to $F_{RA}$ which mixes the transverse parts of the electric and magnetic fields. We define the expansion coefficients as

$$F_{RA} = \sum_{n=0}^{\infty} \frac{1}{R^{n+1}} \overset{n}{F}_{RA} = \sum_{n=0}^{\infty} \sum_{k=0}^{n} \frac{1}{R^{n+1}} U^{n-k} \overset{n,k}{F}_{RA}\Big|_{\mathscr{I}_-^+}. \tag{77}$$

Again, we first set the magnetic multipole moments to zero for simplicity of the presentation. We have

$$F_{RA} = \sum_{n=0}^{\infty} \sum_{\ell=n}^{\infty} \frac{c_{\ell n}}{\ell!} \left( \frac{\overset{(\ell-n)}{Q_L}(U)}{R^{n+1}} \partial_A n_L - \frac{\overset{(\ell-n+1)}{Q_{jL}}(U)\left(n_j \partial_A n_L + (n+1)\partial_A n_j n_L\right)}{R^{n+1}} \frac{1}{(\ell+1)} \right), \tag{78}$$

and thus

$$\overset{n}{F}_{RA} = \sum_{\ell=n}^{\infty} \frac{c_{\ell n}}{\ell!} \left( \overset{(\ell-n)}{Q_L}(U)\partial_A n_L - \overset{(\ell-n+1)}{Q_{jL}}(U)\frac{\left(n_j \partial_A n_L + (n+1)\partial_A n_j n_L\right)}{(\ell+1)} \right). \tag{79}$$

Again, the projection on odd vector harmonics vanishes. We find

$$\langle \overset{n}{F}_{RA}, D^A n_L \rangle = \frac{\mathcal{C}_\ell}{\ell!} \overset{(\ell-n)}{Q_L}(U)\left(\ell(\ell+1)c_{\ell n} - c_{\ell-1,n}(\ell+1)(\ell+n)\right), \tag{80}$$

which vanishes when $n = 0$ as expected from Appendix A.1. From this projection, we find the following matching relations:

$$\overset{n,k}{F}_{RA}(n^i)\Big|_{\mathscr{I}_-^+} = (-1)^{n+1} \overset{n,k}{F}_{RA}(-n^i)\Big|_{\mathscr{I}_+^-}, \qquad \forall\, n \geq 0, \quad 0 \leq k \leq n, \tag{81}$$

obtained from

$$\overset{n,k}{F}_{RA}\Big|_{\mathscr{I}_+^-} = \sum_{j=0}^{\infty} (-2)^j \binom{n+j-k}{j} \left( \overset{n+j,k}{F}_{RA}\Big|_{\mathscr{I}_-^+} - 2\,\overset{n+j+1,k+1}{F}_{UA}\Big|_{\mathscr{I}_-^+} \right). \tag{82}$$

The result extends straightforwardly in the presence of magnetic multipole moments. Note that the factor $(-1)^{n+1}$ in the antipodal matching relation differs by $-1$ compared to the other components of the field strength. This is purely due to us artificially starting the radial expansion at $\mathscr{I}_-^+$ at order $O(R^{-1})$ although $F_{RA}$ is actually zero at that order. While the term $\overset{0}{F}_{RA}$ vanishes at $\mathscr{I}^+$, the analogous term does not vanish at $\mathscr{I}^-$. In order to write a uniform expansion between both null infinities we kept that notation here. This only occurs for this field strength component.

## 3.2 Multipoles as charges

In this section, we will derive the electric multiple moment $Q_L(U)$ as a Noether charge associated with a specific gauge parameter $\varepsilon$. The charge formula in the case of electromagnetism is

$$Q[\varepsilon] = \frac{1}{4}\text{FP} \oint d\Omega \sqrt{-g}\, \varepsilon\, F_{UR}, \tag{83}$$

where usually the finite part FP is the finite value in the limit $R \to \infty$. Here, we will derive a prescription for $\varepsilon$ such that the finite part, which will remove in general diverging terms in the large $R$ expansion, will provide a definition of the electric non-radiative charges $q_{L,k}^+$ extracted from non-radiative fields where the multipolar moments are functions of the non-radiative charges, $Q_L = \sum_{k=0}^{\ell} q_{L,k}^+ U^k$.

First, let us derive the solutions to the wave equation

$$\Box \varepsilon = 0, \tag{84}$$

including those that do not vanish at infinity. We will work in retarded coordinates with metric $ds^2 = -dU^2 - 2\,dU\,dR + R^2 d\Omega^2$ such that $\sqrt{-g} = R^2$. The wave operator can be written

$$\begin{aligned}
\Box \varepsilon &= \frac{1}{\sqrt{-g}} \partial_\mu \left( \sqrt{-g}\, g^{\mu\nu} \partial_\nu \varepsilon \right) \\
&= R^{-2} \left[ \partial_R(R^2 \partial_R \varepsilon) - \partial_R(R^2 \partial_U \varepsilon) - R^2 \partial_U \partial_R \varepsilon + \nabla^2 \varepsilon \right],
\end{aligned} \tag{85}$$

where $\nabla^2$ is the Laplacian on the sphere. We will use the ansatz

$$\varepsilon = R^k f(x) n_L, \qquad x = R/U, \tag{86}$$

in which case the wave equation reduces to

$$(2x^3 + x^2)f''(x) + \left[(2k+4)x^2 + (2k+2)x\right]f'(x) + \left[k(k+1) - \ell(\ell+1)\right]f(x) = 0. \tag{87}$$

The general solutions are given by hypergeometric functions,

$$\begin{aligned}
f_1(x) &= x^{-\ell-k-1}\,{}_2F_1[-\ell-k-1, -\ell, -2\ell, -2x], \\
f_2(x) &= x^{\ell-k}\,{}_2F_1[\ell+1, \ell-k, 2\ell+2, -2x].
\end{aligned} \tag{88}$$

In the large $x$ limit, we have $f_2 = O(x^0)$. We select the solution $f_2$ with $k$ ranging from 0 to $\ell$. For $k = \ell$, we recover the solution $f_2 = 1$ used in [44].

We recall the field for non-radiative configurations,

$$F_{UR} = \sum_{j=0}^{\infty} \sum_{\ell=j}^{\infty} \frac{c_{\ell j}}{(\ell+1)!} \left[ \frac{\overset{(\ell-j+2)}{Q_{L+1}(U)}}{R^{j+1}} n_{L+1} - (\ell+1) \left( \frac{\overset{(\ell-j+1)}{Q_L(U)}}{R^{j+1}} + (j+1) \frac{\overset{(\ell-j)}{Q_L(U)}}{R^{j+2}} \right) n_L \right]. \tag{89}$$

For a given $\ell \geq 0$ and any $k \geq 0$, the charge (83) diverges as $R^{k+1}$. Our prescription is to ignore all divergences and capture the finite part of the expression. After some small algebra, we find it is exactly given by

$$Q[\varepsilon] = f_{\ell,k} \overset{(\ell-k)}{Q_L}(U), \tag{90}$$

where the normalization factor is

$$f_{\ell,k} = \frac{2^{k-\ell-2} k! (2\ell+1)!}{\ell! (2\ell+1)!! (k+\ell-1)!} \left( \Theta_{\ell-k-2} c_{\ell-1,k+1} - \Theta_{\ell-k-1} c_{\ell,k+1} + \Theta_{\ell-k} \frac{k+1}{\ell+1} c_{\ell,k} \right). \tag{91}$$

For $k = \ell$, only the third term is present in $f_{\ell,k}$. For $k = \ell-1$ the second and third term are present while for $0 \leq k \leq \ell-2$ all terms are present. It is obvious from the charge expression (90) that using the $\ell+1$ integer values of $k$ in the range $0 \leq k \leq \ell$, we can deduce the values of all non-radiative electric multipoles $q_{L,k}^+$ for a given $\ell$.

## 3.3 Radiative case: Antipodal matchings for logarithmic corrections at $\mathscr{I}^\pm$

The authors of [30] postulated a matching relation between logarithmic contributions to the field strength between $\mathscr{I}^+$ and $\mathscr{I}^-$. The postulated matching relation is expressed as

$$\overset{0,\ln U}{F^-_{RA}}(n^i)\Big|_{\mathscr{I}^+} = \overset{0,\ln V}{F^+_{RA}}(-n^i)\Big|_{\mathscr{I}^-}, \tag{92}$$

where the field strength was expanded as

$$F_{RA} = \frac{\ln R}{R^2} \overset{\ln}{F}_{RA} + \frac{1}{R^2} \overset{0}{F}_{RA} + \dots, \tag{93}$$

with

$$\begin{aligned}
\overset{0}{F}_{RA}(U, n^i)\big|_{\mathscr{I}^+} &\overset{U \to \pm\infty}{=} U \overset{1,0}{F^{\pm}}_{RA}(n^i) + \ln U \overset{0,\ln U}{F^{\pm}}_{RA}(n^i) + O(U^0), \\
\overset{0}{F}_{RA}(V, n^i)\big|_{\mathscr{I}^-} &\overset{V \to \pm\infty}{=} V \overset{1,0}{F^{\pm}}_{RA}(n^i) + \ln V \overset{0,\ln V}{F^{\pm}}_{RA}(n^i) + O(V^0).
\end{aligned} \tag{94}$$

Importantly, we have shown that the presence of $\ln U$ terms at $\mathscr{I}^+$ does not imply the existence of $\ln V$ terms at $\mathscr{I}^-$, such that (92) does not make sense in our view.

Instead, we show that antipodal matching relations exist between $\ln U$ terms at $\mathscr{I}^+$ and $\ln R$ terms at $\mathscr{I}^-$, in agreement with the findings of Bhaktar [23,38]. First, note that Eqs. (56), (64), (74) and (80) do not depend on the nature of the multipole moment. As shown in Section 2.2, radiation induces, at null infinity and for each multipole moment $Q_L(U)$ with $\ell \geq 1$, an infinite number of new terms at order $O(e^3)$ dominated by $\ln|U|U^{\ell-1}$. In Appendix A.2, we state all the expansions for the $O(e^3)$ contributions to the field strength at $\mathscr{I}^+$ and how they translate at $\mathscr{I}^-$. Given that the transformation laws from the $\ln U$ to $\ln R$ terms are the same as the ones for the non-radiative contributions, we can immediately infer that antipodal matching relations will also exist between these terms. For instance, for the radial electric field $F_{UR}$, we recall that that the projection on arbitrary harmonics $n_L$ is given by Eq. (56). Plugging the dominant logarithmic contribution of Eq. (52), we get

$$\langle \overset{n+1,\ln}{F}_{UR}, n_L \rangle = \frac{\mathcal{C}_{\ell}}{\ell n!} q^{+,(\ln)} \left( c_{\ell-1,n+2} - c_{\ell,n+2} - (n+2)c_{\ell,n+1} \right). \tag{95}$$

Up to a $\ell$-dependent factor, this has the same exact form as the tree-level expression (56) where we shift $n \mapsto n+1$ and we set $k = 0$. From this, we find

$$\overset{n+1,\ln}{F}_{UR}(n^i)\big|_{\mathscr{I}^+_-} = (-1)^{n+1} \overset{n+1,\ln}{F}_{VR}(-n^i)\big|_{\mathscr{I}^-_+}. \tag{96}$$

Likewise, we get antipodal matchings for the other logarithmic contributions to the field strength,

$$\overset{n+1,\ln}{F}_{UA}(n^i)\big|_{\mathscr{I}^+_-} = (-1)^{n+1} \overset{n+1,\ln}{F}_{VA}(-n^i)\big|_{\mathscr{I}^-_+}, \tag{97}$$

$$\overset{n+1,\ln}{F}_{AB}(n^i)\big|_{\mathscr{I}^+_-} = (-1)^{n+1} \overset{n+1,\ln}{F}_{AB}(-n^i)\big|_{\mathscr{I}^-_+}, \tag{98}$$

and

$$\overset{n+1,\ln}{F}_{RA}(n^i)\big|_{\mathscr{I}^+_-} = (-1)^{n} \overset{n+1,\ln}{F}_{RA}(-n^i)\big|_{\mathscr{I}^-_+}, \tag{99}$$

where we remind the reader that quantities at $\mathscr{I}^+_-$ depend logarithmically on U, see Eqs. (A.21), (A.26), (A.29) and (A.32) while those at $\mathscr{I}^-_+$ depend logarithmically on $R$, see Eqs. (A.24), (A.27), (A.30) and (A.33).

Let us note that the antipodal relation (99) for $n = 0$ had been first proposed in [38] as a way to reproduce the logarithmic soft photon theorem, as discussed in the next section, and subsequently proved to hold for the electromagnetic field produced by a set of charged point particles in [23]. Here we have shown that there exist such antipodal relations in all components of the electromagnetic field and to all orders in $R$. Moreover, our derivation does not rely on a particular set of solutions such as those associated with charged point particles, but rather on the validity of the asymptotic expansion (51). This broadens the applicability of the antipodal matching relations.

# 4 Logarithmic soft photon theorem

As a direct application of the antipodal matching relations obtained in the previous section, we turn to the derivation of the logarithmic soft photon theorem in its classical form. While the antipodal matching relation (92) used as a starting point for the derivation in [30] is incorrect, a substantial part of their analysis still applies. After deriving our results, we got aware of the work [38] which derives the logarithmic soft photon theorem from the correct antipodal matching relation, which was conjectured in that work. Our derivation therefore complements the alternative derivation of [38] based on different methods.

We start from the antipodal matching relation (99) with $n = 0$, namely

$$\overset{1,\ln}{F}_{RA}(n)\big|_{\mathscr{I}^+_-} = \overset{1,\ln}{F}_{RA}(-n)\big|_{\mathscr{I}^-_+}, \tag{100}$$

which we integrate over the sphere against some arbitrary vector field $V^A(n)$,

$$Q_+[V] \equiv -\oint V^A(n)\overset{1,\ln}{F}_{RA}(n)\big|_{\mathscr{I}^+_-} = -\oint V^A(n)\overset{1,\ln}{F}_{RA}(-n)\big|_{\mathscr{I}^-_+} \equiv Q_-[V], \tag{101}$$

where we keep the conventional minus sign used in [30]. We will discuss $Q_+$ and $Q_-$ separately, with the analysis of $Q_+$ following closely that of [30]. In particular, using the asymptotic expansion (C.4) of the Maxwell field, we can write

$$\begin{aligned}
\overset{1,\ln}{F}_{RA}\big|_{\mathscr{I}^+_-} &= -\lim_{U \to -\infty} U^2 \partial_U^2 \overset{1}{F}_{RA} = \int_{-\infty}^{\infty} dU\, \partial_U[U^2 \partial_U^2 \overset{1}{F}_{RA}] - \lim_{U \to \infty} U^2 \partial_U^2 \overset{1}{F}_{RA} \\
&= \int_{-\infty}^{\infty} dU\, \partial_U[U^2 \partial_U^2 \overset{1}{F}_{RA}] + \overset{1,\ln}{F}_{RA}\big|_{\mathscr{I}^+_+},
\end{aligned} \tag{102}$$

where assumed that the expansion in $U \to \infty$ towards $\mathscr{I}^+_+$ has a structure similar to that at $\mathscr{I}^+_-$. Then using the equation of motion (C.13) determining $\partial_U \overset{1}{F}_{RA}$, we have

$$\overset{1,\ln}{F}_{RA}\big|_{\mathscr{I}^+_-} = \int_{-\infty}^{\infty} dU\, \partial_U[U^2 \partial_U[\overset{1}{F}_{UA} - D^B \overset{0}{F}_{BA} - q^2 \overset{0}{j}_A]] + \overset{1,\ln}{F}_{RA}\big|_{\mathscr{I}^+_+}. \tag{103}$$

Here we will assume a scattering of massive charged particles such that the charged current $j_A$ at $\mathscr{I}^+$ can be set to zero. On the other hand we use the $U$-expansion (C.2) of $\overset{1}{F}_{UA}$ to show that this term does not actually contribute to the integral, such that

$$\overset{1,\ln}{F}_{RA}\big|_{\mathscr{I}^+_-} = -\int_{-\infty}^{\infty} dU\, \partial_U[U^2 \partial_U D^B \overset{0}{F}_{BA}] + \overset{1,\ln}{F}_{RA}\big|_{\mathscr{I}^+_+}. \tag{104}$$

This allows to split the charge $Q_+[V]$ into two contributions,

$$Q_+[V] = Q_+^{\text{soft}}[V] + Q_+^{\text{hard}}[V], \tag{105}$$

with

$$\begin{aligned}
Q_+^{\text{soft}}[V] &\equiv \int_{\mathscr{I}^+} V^A \partial_U[U^2 \partial_U D^B \overset{0}{F}_{BA}], \\
Q_+^{\text{hard}}[V] &\equiv -\oint V^A \overset{1,\ln}{F}_{RA}\big|_{\mathscr{I}^+_+}.
\end{aligned} \tag{106}$$

In [30] it is shown that the hard charge can be written in terms of the logarithmic component of the charged matter current $j_\alpha$ on the 'blow-up' hyperboloid $\mathbb{H}_3$ of future timelike infinity $i^+$, namely

$$Q_+^{\text{hard}}[V] = -\int_{\mathbb{H}_3} V^\alpha(y) \overset{\ln}{j_\alpha}(y), \tag{107}$$

where the vector field $V(y) = V^\alpha(y) dy_\alpha$ on the hyperboloid $\mathbb{H}_3$ is explicitly given by

$$V^\alpha(y) = \frac{1}{8\pi} \oint (-q(n) \cdot v(y))^{-3} J_{\mu\nu}^\alpha(y) L_A^{\mu\nu}(n) V^A(n). \tag{108}$$

Here $q^\mu = (1, n^i)$ and $v^\mu(y)$ are embeddings of the celestial sphere and of the hyperboloid $\mathbb{H}_3$ into Minkowski space $\mathbb{M}$, satisfying in particular $q^2 = 0$ and $v^2 = -1$. The quantities $J_{\mu\nu}^\alpha(y)$ and $L_A^{\mu\nu}(n)$ are representations of the Lorentz generators on $\mathbb{H}_3$ and on the celestial sphere, respectively given by

$$\begin{aligned} L_A^{\mu\nu} &= q^\mu \partial_A q^\nu - q^\nu \partial_A q^\mu, \\ J_\alpha^{\mu\nu} &= v^\mu \partial_\alpha v^\nu - v^\nu \partial_\alpha v^\mu. \end{aligned} \tag{109}$$

In Fourier space, the soft charge can be written [30]

$$Q_+^{\text{soft}}[V] = \lim_{\omega \to 0} \partial_\omega \omega^2 \partial_\omega \oint V^A(n) D_A D^B \overset{0}{A}_B(\omega, n), \tag{110}$$

in terms of the asymptotic photon field $A_B^0(\omega, n)$. Furthermore the authors of [30] show that upon quantization, the hard charge evaluated in a state $|\text{out}\rangle$ consisting of a set of outgoing massive charged particles is given by

$$Q_+^{\text{hard}}[V] |\text{out}\rangle = \sum_{a,b \in \text{out}} \frac{e_a^2 e_b}{4\pi m_a} V^\alpha(y_a) \frac{\partial}{\partial y_a^\alpha} \left( \frac{v_a \cdot v_b}{\sqrt{(v_a \cdot v_b)^2 - 1}} \right) |\text{out}\rangle, \tag{111}$$

where we recall that the velocities $v_a(y_a)$ are implicitly parametrized by the coordinates $y_a^\alpha$. Indeed the blow-up hyperboloid $\mathbb{H}_3$ also plays the role of (normalized) momentum mass-shell for the massive particle states [45].

Now we turn to the analysis of $Q_-[V]$, which departs entirely from that presented in [30], although its final expression will be the desired one. For a set of massive charged particles incoming from past timelike infinity $i^-$, we have explicitly computed in (32) the corresponding field component

$$\overset{1,\ln}{F}_{RA}(n^i)\big|_{\mathscr{I}_+^-} = \sum_{a \in \text{in}} \frac{e_a}{4\pi} \frac{e_A^\mu \left( c_\mu^a v_\nu^a - c_\nu^a v_\mu^a \right) n^\nu}{(n \cdot v_a)^3}, \tag{112}$$

where we recall that $n_\mu = (1, n_i)$ and $e_A^\mu = (0, e_A^i)$. Thus evaluation at the antipodal point instead gives

$$\overset{1,\ln}{F}_{RA}(-n^i)\big|_{\mathscr{I}_+^-} = \sum_{a \in \text{in}} \frac{e_a}{4\pi} \frac{\partial_A q^\mu \left( c_\mu^a v_\nu^a - c_\nu^a v_\mu^a \right) q^\nu}{(-q \cdot v_a)^3}. \tag{113}$$

With this let us prove that $Q_-[V]$ as defined in (101) also equals the analogue of the hard charge (111) for ingoing particles, namely

$$Q_-^{\text{hard}}[V] \equiv \sum_{a,b \in \text{in}} \frac{e_a^2 e_b}{4\pi m_a} V^\alpha(y_a) \frac{\partial}{\partial y_a^\alpha} \left[ \frac{v_a \cdot v_b}{\sqrt{(v_a \cdot v_b)^2 - 1}} \right]. \tag{114}$$

To show this, let us first insert (108) in this expression, yielding

$$Q_-^{\text{hard}}[V] = \sum_{a,b \in \text{in}} \frac{e_a^2 e_b}{32\pi^2 m_a} \oint (-q \cdot v_a)^{-3} V^A(n) L_A^{\mu\nu}(n) J_{\mu\nu}^\alpha(y_a) \frac{\partial}{\partial y_a^\alpha}\left[ \frac{v_a \cdot v_b}{\sqrt{(v_a \cdot v_b)^2 - 1}} \right]. \quad (115)$$

We have

$$J_{\mu\nu}^\alpha(y_a) \frac{\partial}{\partial y_a^\alpha}\left[ \frac{v_a \cdot v_b}{\sqrt{(v_a \cdot v_b)^2 - 1}} \right] = -\frac{J_{\mu\nu}^\alpha(y_a) \partial_\alpha v_a \cdot v_b}{[(v_a \cdot v_b)^2 - 1]^{3/2}} = -\frac{v_\mu^a v_\nu^b - v_\nu^a v_\mu^b}{[(v_a \cdot v_b)^2 - 1]^{3/2}}, \quad (116)$$

where the last equation follows from the resolution of the identity

$$\partial_\alpha v_\mu \, \partial^\alpha v_\nu = \eta_{\mu\nu} + v_\mu v_\nu. \quad (117)$$

Hence we can write

$$\begin{aligned}
Q_-^{\text{hard}}[V] &= -\sum_{a,b \in \text{in}} \frac{e_a^2 e_b}{32\pi^2 m_a} \oint (-q \cdot v_a)^{-3} V^A(n) L_A^{\mu\nu}(n) \frac{v_\mu^a v_\nu^b - v_\nu^a v_\mu^b}{[(v_a \cdot v_b)^2 - 1]^{3/2}} \\
&= \sum_{a \in \text{in}} \frac{e_a}{8\pi} \oint (-q \cdot v_a)^{-3} V^A(n) L_A^{\mu\nu}(n)(v_\mu^a c_\nu^a - v_\nu^a c_\mu^a) \\
&= \sum_{a \in \text{in}} \frac{e_a}{4\pi} \oint V^A(n) \frac{\partial_A q^\mu (c_\mu^a v_\nu^a - c_\nu^a v_\mu^a) q^\nu}{(-q \cdot v_a)^3} = \oint V^A(n) \overset{1,\ln}{F}_{RA}(-n^i)\big|_{\mathscr{I}_+^-},
\end{aligned} \quad (118)$$

recovering the original expression (101), and thereby establishing $Q_-[V] = Q_-^{\text{hard}}[V]$.

We now have everything at our disposal to derive the classical part of the logarithmic soft photon theorem. Indeed, assuming that the scattering dynamics encoded in the $\mathcal{S}$-matrix does not violate the conservation of the charge $Q[V] = Q_+[V] = Q_-[V]$, we can write

$$[Q[V], \mathcal{S}] = 0, \quad (119)$$

or equivalently

$$\langle \text{out}| Q_+^{\text{soft}}[V] \mathcal{S} |\text{in}\rangle = \langle \text{out}| \mathcal{S} Q_-^{\text{hard}}[V] |\text{in}\rangle - \langle \text{out}| Q_+^{\text{hard}}[V] \mathcal{S} |\text{in}\rangle. \quad (120)$$

With the expressions of the charges given in Eqs. (110)-(111) and Eq. (114), it was shown by the authors of [30] that the identity (120) is nothing but a rewriting of the classical logarithmic soft photon theorem as originally presented in [18].

In summary, the systematic study initiated here has revealed new sets of antipodal matching relations at tree level (60)-(68)-(75)-(81), and at first order in matter-field interactions (97)-(98)-(99), which were mostly previously unknown. We have confirmed and broadened the applicability of the antipodal relation (99) for $n = 0$ that had been put forward in [23,38], and which underlies the logarithmic soft photon theorem. Given that these antipodal matching relations control multipoles sitting at all orders in the large-$R$ and large-$U$ expansions, we can expect that they control sub$^n$-leading soft photon theorems to all orders in the soft expansion, such as those discussed in [22,46–49]. We leave this exciting prospect to future investigations.

## Acknowledgments

**Funding information** GC is Research Director of the FNRS. DF benefits from a FRIA Fellowship from the FNRS. KN is supported by a Postdoctoral Fellowship granted by the FNRS.

# A  Multipole expansions for the electromagnetic fields

In this section, we describe the most general form of each component of the electromagnetic field, first at spatial infinity, i.e. in a non-radiative setting, and then when radiation is included. The multipole moments we consider are given by Eq. (51), and their derivatives by Eq. (52). In retarded spherical coordinates $(U, R, X^A)$, we have

$$\partial_i A_0(T, X^i) = \sum_{\ell=0}^{\infty} \sum_{j=0}^{\ell} \frac{c_{\ell j}}{\ell!} \left[ \left( \frac{\overset{(\ell-j+1)}{Q_L}(U)}{R^{j+1}} + (j+\ell+1) \frac{\overset{(\ell-j)}{Q_L}(U)}{R^{j+2}} \right) n_i n_L - \ell \frac{\overset{(\ell-j)}{Q_L}(U)}{R^{j+2}} \delta_{i(i_1} n_{i_2} ... n_{i_\ell)} \right], \tag{A.1}$$

$$\partial_T A_i(T, X^i) = \sum_{\ell=0}^{\infty} \sum_{j=0}^{\ell} \frac{c_{\ell j}}{(\ell+1)!} \left[ \frac{\overset{(\ell-j+2)}{Q_{iL}}(U)}{R^{j+1}} - \frac{\ell+1}{\ell+2} \varepsilon_{iab} \left( \frac{\overset{(\ell-j+2)}{M_{bL}}(U)}{R^{j+1}} + (j+1+\ell) \frac{\overset{(\ell-j+1)}{M_{bL}}(U)}{R^{j+2}} \right) n_a \right] n_L, \tag{A.2}$$

and

$$\partial_i A_j(T, X^i) = \sum_{\ell=0}^{\infty} \sum_{k=0}^{\ell} \frac{c_{\ell k}}{(\ell+1)!} \left[ -\left( \frac{\overset{(\ell-k+2)}{Q_{jL}}(U)}{R^{k+1}} + (k+\ell+1) \frac{\overset{(\ell-k+1)}{Q_{jL}}(U)}{R^{k+2}} \right) n_i n_L \right. \tag{A.3}$$

$$- \frac{(\ell+1)^2}{\ell+2} \varepsilon_{jab} \left( \frac{\overset{(\ell-k+1)}{M_{bL}}(U)}{R^{k+2}} + (k+\ell+1) \frac{\overset{(\ell-k)}{M_{bL}}(U)}{R^{k+3}} \right) \delta_{i(a} n_{L)} + \ell \frac{\overset{(\ell-k+1)}{Q_{jL}}(U)}{R^{k+2}} \delta_{i(i_1} n_{i_2} ... n_{i_\ell)}$$

$$+ \frac{\ell+1}{\ell+2} \varepsilon_{jab} n_i n_a n_L \left( \frac{\overset{(\ell-k+2)}{M_{bL}}(U)}{R^{k+1}} + (2k+2\ell+3) \frac{\overset{(\ell-k+1)}{M_{bL}}(U)}{R^{k+2}} + (k+\ell+1)(k+\ell+3) \frac{\overset{(\ell-k)}{M_{bL}}(U)}{R^{k+3}} \right) \right].$$

From these, we find that the radial electric field is given by

$$F_{UR} = \frac{\partial X^\mu}{\partial U} \frac{\partial X^\nu}{\partial R} F_{\mu\nu} = n^i F_{Ti}$$
$$= \sum_{\ell=0}^{\infty} \sum_{j=0}^{\ell} \frac{c_{\ell j}}{(\ell+1)!} \left[ \frac{\overset{(\ell-j+2)}{Q_{L+1}}(U)}{R^{j+1}} n_{L+1} - (\ell+1) \left( \frac{\overset{(\ell-j+1)}{Q_L}(U)}{R^{j+1}} + (j+1) \frac{\overset{(\ell-j)}{Q_L}(U)}{R^{j+2}} \right) n_L \right], \tag{A.4}$$

the transverse electric field by

$$F_{UA} = \frac{\partial X^\mu}{\partial U} \frac{\partial X^\nu}{\partial X^A} F_{\mu\nu} = R e_A^i F_{Ti}$$
$$= e_A^i \sum_{j=0}^{\infty} \sum_{\ell=j}^{\infty} \frac{c_{\ell j}}{(\ell+1)!} \left[ \frac{\overset{(\ell-j+2)}{Q_{iL}}(U)}{R^j} - \frac{\ell+1}{\ell+2} \varepsilon_{iab} \left( \frac{\overset{(\ell-j+2)}{M_{bL}}(U)}{R^j} + (j+1+\ell) \frac{\overset{(\ell-j+1)}{M_{bL}}(U)}{R^{j+1}} \right) n_a \right] n_L \tag{A.5}$$

$$+ \sum_{j=0}^{\infty} \sum_{\ell=j}^{\infty} \frac{c_{\ell j}}{(\ell+1)!} \ell(\ell+1) \frac{\overset{(\ell-j)}{Q_L}(U)}{R^{j+1}} e_{(i_1}^A n_{i_2} ... n_{i_\ell)},$$

and the sum of the transverse components of the electric and magnetic field as

$$
\begin{aligned}
F_{RA} &= \frac{\partial X^\mu}{\partial R}\frac{\partial X^\nu}{\partial X^A}F_{\mu\nu} = F_{UA} + Rn^i e_A^j F_{ij} \\
&= \sum_{k=0}^\infty \sum_{\ell=k}^\infty \frac{c_{\ell k}}{\ell!}\left[ \frac{\overset{(\ell-k)}{Q_L}(U)}{R^{k+1}}\partial_A n_L - \frac{\overset{(\ell-k+1)}{Q_{jL}}(U)\left(n_j\partial_A n_L + (k+1)\partial_A n_j n_L\right)}{R^{k+1}}\frac{}{(\ell+1)}\right. \\
&\qquad \left. + \frac{1}{(\ell+2)}\epsilon_{jab}\left(k\frac{\overset{(\ell-k+1)}{M_{bL}}(U)}{R^{k+1}} + (k+1)(k+\ell+1)\frac{\overset{(\ell-k)}{M_{bL}}(U)}{R^{k+2}}\right)e_A^j n_a n_L\right].
\end{aligned}
\tag{A.6}
$$

Finally, we compute the expansion for the magnetic field projected on the sphere. Because the multipole moments are totally symmetric, we find that only the magnetic multipoles contribute to this component:

$$
\begin{aligned}
F_{AB} &= \frac{\partial X^\mu}{\partial X^A}\frac{\partial X^\nu}{\partial X^B}F_{\mu\nu} = R^2 e_A^i e_B^j F_{ij} \\
&= \sum_{k=0}^\infty \sum_{\ell=k}^\infty \frac{c_{\ell k}}{\ell!(\ell+2)}\epsilon_{jab}\, e_A^j\left(e_B^a n_L + n_a\partial_B n_L\right)\left(\frac{\overset{(\ell-k+1)}{M_{bL}}(U)}{R^k} + (k+\ell+1)\frac{\overset{(\ell-k)}{M_{bL}}(U)}{R^{k+1}}\right).
\end{aligned}
\tag{A.7}
$$

## A.1 Non-radiative expansions at $i^0$

In this subsection, equipped with the exact expressions (A.4)-(A.7) for the electromagnetic fields, we consider only the first, polynomial, contribution to Eq. (52). Let us introduce the index $d$ defined as the difference between the multipole index of $Q_L(U)$ and the order of differentiation we apply to this multipole. At spatial infinity, any contribution with $d < 0$ vanishes, and $d \geq 0$ will involve polynomials of order $d$. From the expansions (A.4)-(A.7), we see that this index $d$ increases by one with every decreasing order in the radial coordinate $R$. Then, the whole $(U,R)$ structure of the electromagnetic field components at spatial infinity is determined by the index $d$ of the leading radial order terms.

**Radial electric field** Although Eq. (A.4) formally involves $O(R^{-1})$ terms, one can observe that all the multipoles contributing to this order have $d = -1$ and thus vanish. As explained above, the first non-vanishing order will carry an index $d = 0$, and so on. We find that, at spatial infinity, the radial electric field can be written as

$$
F_{UR} = \sum_{n=0}^\infty \frac{1}{R^{n+2}}\overset{n}{F}_{UR} = \sum_{n=0}^\infty \sum_{k=0}^n \frac{1}{R^{n+2}}U^{n-k}\overset{n,k}{F}_{UR}.
\tag{A.8}
$$

When going to advanced spherical coordinates with $U = V - 2R$, the field strength transforms as $F_{VR}(V,R,X^A) = F_{UR}(U(V,R,X^A),R,X^A)$ and

$$
F_{VR} = \sum_{n=0}^\infty \frac{1}{R^{n+2}}\overset{n}{F}_{VR} = \sum_{n=0}^\infty \sum_{k=0}^n \frac{1}{R^{n+2}}V^{n-k}\overset{n,k}{F}_{VR},
\tag{A.9}
$$

with

$$
\overset{n,k}{F}_{VR} = \sum_{j=0}^\infty (-2)^j \binom{n+j-k}{j}\overset{n+j,k}{F}_{UR}.
\tag{A.10}
$$

Indeed,

$$
\begin{aligned}
\sum_{n=0}^{\infty}\sum_{k=0}^{n}\frac{1}{R^{n+2}}U^{n-k}\overset{n,k}{F}_{UR} &= \sum_{n=0}^{\infty}\sum_{k=0}^{n}\sum_{j=0}^{n-k}\frac{1}{R^{n-j+2}}(-2)^{j}V^{n-k-j}\binom{n-k}{j}\overset{n,k}{F}_{UR}\\
&= \sum_{n=0}^{\infty}\sum_{k=0}^{n}\sum_{j=-n}^{k}\frac{1}{R^{-j+2}}(-2)^{j+n}V^{-k-j}\binom{n-k}{j+n}\overset{n,k}{F}_{UR}\\
&= \sum_{n=0}^{\infty}\sum_{k=0}^{n}\sum_{j=k}^{n}\frac{1}{R^{j+2}}(-2)^{n-j}V^{j-k}\binom{n-k}{n-j}\overset{n,k}{F}_{UR}\\
&= \sum_{n=0}^{\infty}\sum_{j=0}^{n}\sum_{k=0}^{j}\cdots = \sum_{j=0}^{\infty}\sum_{n=j}^{\infty}\sum_{k=0}^{j}\cdots\\
&= \sum_{n=0}^{\infty}\sum_{k=0}^{n}\frac{1}{R^{n+2}}V^{n-k}\sum_{j=0}^{\infty}(-2)^{j}\binom{n+j-k}{j}\overset{n+j,k}{F}_{UR},
\end{aligned}
$$

where, in the first and last lines, we respectively use Leibniz's differentiation rule and perform the successive transformations $n \to n-j$ and $n \leftrightarrow j$. Although the sum over $j$ seemingly sweeps all the positive integers, it is naturally truncated by the definition of the $c_{\ell m}$ coefficients introduced in Eq. (3), which vanish for $m > \ell$.

**Transverse electric field** The first non-vanishing order in the radial coordinate is $O(R^{-1})$ with $d = 0$. Then,

$$
F_{UA} = \sum_{n=0}^{\infty}\frac{1}{R^{n+1}}\overset{n+1}{F}_{UA} = \sum_{n=0}^{\infty}\sum_{k=0}^{n}\frac{1}{R^{n+1}}U^{n-k}\overset{n+1,k+1}{F}_{UA}, \tag{A.11}
$$

where the $n+1$ and $k+1$ superscripts are chosen for notational convenience, as will be apparent in the last section. When going to advanced coordinates, the field strength transforms trivially as $F_{VA}(V,R,X^A) = F_{UA}(U(V,R,X^A),R,X^A)$ and

$$
F_{VA} = \sum_{n=0}^{\infty}\frac{1}{R^{n+1}}\overset{n+1}{F}_{VA} = \sum_{n=0}^{\infty}\sum_{k=0}^{n}\frac{1}{R^{n+1}}V^{n-k}\overset{n+1,k+1}{F}_{VA}, \tag{A.12}
$$

with

$$
\overset{n+1,k+1}{F}_{VA} = \sum_{j=0}^{\infty}(-2)^{j}\binom{n+j-k}{j}\overset{n+j+1,k+1}{F}_{UA}. \tag{A.13}
$$

**$F_{RA}$ component** One can check that the $O(R^{-1})$ contribution to $F_{RA}$ vanishes for any multipole moment: we have

$$
F_{RA} = \frac{1}{R}\sum_{\ell=0}^{\infty}\frac{1}{\ell!}\left(\overset{(\ell)}{Q}_{L}(U)\partial_{A}n_{L} - \frac{1}{\ell+1}\overset{(\ell+1)}{Q}_{L+1}(U)\partial_{A}n_{L+1}\right) + O(R^{-2}). \tag{A.14}
$$

By shifting the summation index in the second term, we find that the two $O(R^{-1})$ contributions cancel each other, and the first non-trivial order has $d = 1$. Nevertheless, in order to simplify our analysis of the matching relations by homogenising the expansions at future and past null infinities, let us write

$$
F_{RA} = \sum_{n=0}^{\infty}\frac{1}{R^{n+1}}\overset{n}{F}_{RA} = \sum_{n=0}^{\infty}\sum_{k=0}^{n}\frac{1}{R^{n+1}}U^{n-k}\overset{n,k}{F}_{RA}\Big|_{\mathscr{I}_{-}^{+}}, \tag{A.15}
$$

where we acknowledge that $\overset{0,0}{F_{RA}}\big|_{\mathscr{I}^+}$ vanishes. Going to advanced coordinates, the Jacobian matrix is non-trivial: $F_{RA}(V,R,X^A) = F_{RA}(U(V,R),R,X^A) - 2F_{UA}(U(V,R),R,X^A)$ and $F_{RA}$ can be expanded as

$$F_{RA} = \sum_{n=0}^{\infty}\sum_{k=0}^{n}\frac{1}{R^{n+1}}V^{n-k}\overset{n,k}{F_{RA}}\Bigg|_{\mathscr{I}^-_+}, \tag{A.16}$$

with

$$\overset{n,k}{F_{RA}}\Big|_{\mathscr{I}^-_+} = \sum_{j=0}^{\infty}(-2)^j\binom{n+j-k}{j}\left(\overset{n+j,k}{F_{RA}}\Big|_{\mathscr{I}^+_-} - 2\overset{n+j+1,k+1}{F_{UA}}\Big|_{\mathscr{I}^+_-}\right). \tag{A.17}$$

Note that $F_{RA}$ has a non-trivial $O(R^{-1})$ contribution in advanced coordinates.

**Radial magnetic field**  From (A.7), we straightforwardly find

$$F_{AB} = \sum_{n=0}^{\infty}\frac{1}{R^n}\overset{n}{F_{AB}} = \sum_{n=0}^{\infty}\sum_{k=0}^{n}\frac{1}{R^n}U^{n-k}\overset{n,k}{F_{AB}}\big|_{\mathscr{I}^+_-}, \tag{A.18}$$

and

$$F_{AB} = \sum_{n=0}^{\infty}\frac{1}{R^n}\overset{n}{F_{AB}} = \sum_{n=0}^{\infty}\sum_{k=0}^{n}\frac{1}{R^{n+1}}V^{n-k}\overset{n,k}{F_{AB}}\big|_{\mathscr{I}^-_+}, \tag{A.19}$$

with

$$\overset{n,k}{F_{AB}}\Big|_{\mathscr{I}^-_+} = \sum_{j=0}^{\infty}(-2)^j\binom{n+j-k}{j}\overset{n,k}{F_{AB}}\Big|_{\mathscr{I}^+_-}. \tag{A.20}$$

## A.2 Field expansions at $\mathscr{I}^\pm$: Radiative corrections

We now turn to the $O(e^3)$ corrections to the multipole moments given in (52).

**Radial electric field**  From (52) and (A.4), we find the $O(e^3)$ contributions to the radial electric field to be given by

$$F_{UR} = \sum_{n=0}^{\infty}\frac{1}{R^{n+3}}\ln|U|U^n\overset{n+1,\text{ln}}{F_{UR}} + \sum_{n=0}^{\infty}\frac{1}{R^{n+2}}O(U^{n-1}), \tag{A.21}$$

where the $\overset{n+1,\text{ln}}{F_{UR}}$ coefficients are generated by the logarithmic part of (51). We now go to advanced coordinates. Note that

$$\ln|U| = \ln(2R-V) = \ln R + \ln 2 - \sum_{m=1}^{\infty}\frac{1}{m}\left(\frac{V}{2R}\right)^m, \tag{A.22}$$

and

$$U^{-m} = \frac{(-1)^m}{2^mR^m}\sum_{k=0}^{\infty}\binom{m+k-1}{k}\left(\frac{V}{2R}\right)^m. \tag{A.23}$$

Then, at $\mathscr{I}^-$, we find

$$F_{VR} = \sum_{n=0}^{\infty}\frac{1}{R^{n+3}}\ln R\,V^n\overset{n+1,\text{ln}}{F_{VR}} + \sum_{n=0}^{\infty}\frac{1}{R^{n+3}}O(V^n), \tag{A.24}$$

with

$$\overset{n+1,\text{ln}}{F_{VR}} = \sum_{j=0}^{\infty}\binom{n+j}{j}(-2)^j\overset{n+j+1,\text{ln}}{F_{UR}}. \tag{A.25}$$

Notice that the transformation law for the angular coefficients of the logarithmic terms is the same as for the tree-level contributions, where $k=0$.

**Transverse electric field**   Performing the same steps as for $F_{UR}$, we find

$$F_{UA} = \sum_{n=0}^{\infty} \frac{1}{R^{n+2}} \ln|U| U^n \overset{n+2,\ln}{F_{UA}} + \sum_{n=0}^{\infty} \frac{1}{R^n} O(U^{n-2}), \tag{A.26}$$

at $\mathscr{I}^+$. Radiation induces overleading contributions to $F_{UA}$ in the radial coordinate. However, these contributions carry a negative power of $U$ and therefore will be strongly suppressed in the neighbourhood of spatial infinity. At $\mathscr{I}^-$, we find

$$F_{VA} = \sum_{n=0}^{\infty} \frac{1}{R^{n+2}} \ln R\, V^n \overset{n+2,\ln}{F_{VA}} + \sum_{n=0}^{\infty} \frac{1}{R^{n+2}} O(V^n), \tag{A.27}$$

with

$$\overset{n+2,\ln}{F_{VA}} = \sum_{j=0}^{\infty} \binom{n+j}{j} (-2)^j \overset{n+j+2,\ln}{F_{UA}}. \tag{A.28}$$

**$F_{RA}$ component**   At order $\mathscr{I}^+$, at order $O(e^3)$, we have

$$F_{RA} = \sum_{n=0}^{+\infty} \frac{1}{R^{n+2}} \ln|U| U^n \overset{n+1,\ln}{F_{RA}}\Big|_{\mathscr{I}^+_-} + \sum_{n=0}^{\infty} \frac{1}{R^{n+2}} O(U^n). \tag{A.29}$$

At $\mathscr{I}^-$, this translates to

$$F_{RA} = \sum_{n=0}^{\infty} \frac{1}{R^{n+2}} \ln R\, V^n \overset{n+1,\ln}{F_{RA}}\Big|_{\mathscr{I}^-_+} + \sum_{n=0}^{\infty} \frac{1}{R^{n+2}} O(V^n), \tag{A.30}$$

with

$$\overset{n+1,\ln}{F_{RA}}\Big|_{\mathscr{I}^-_+} = \sum_{j=0}^{\infty} \binom{n+j}{j} (-2)^j \left( \overset{n+j+1,\ln}{F_{RA}}\Big|_{\mathscr{I}^+_-} - 2 \overset{n+j+2,\ln}{F_{UA}}\Big|_{\mathscr{I}^+_-} \right). \tag{A.31}$$

**Radial magnetic field**   Finally, we find that the subleading corrections to the radial magnetic field can be written as

$$F_{AB} = \sum_{n=0}^{\infty} \frac{1}{R^{n+1}} \ln|U| U^n \overset{n+1,\ln}{F_{AB}}\Big|_{\mathscr{I}^+_-} + \sum_{n=0}^{\infty} \frac{1}{R^n} O(U^{n-1}). \tag{A.32}$$

Thus, we find the following expansion at $\mathscr{I}^-$ for $F_{AB}$:

$$F_{AB} = \sum_{n=0}^{\infty} \frac{1}{R^{n+1}} \ln R\, V^n \overset{n+1,\ln}{F_{AB}}\Big|_{\mathscr{I}^-_+} + \sum_{n=0}^{\infty} \frac{1}{R^{n+1}} O(V^n), \tag{A.33}$$

with

$$\overset{n+1,\ln}{F_{AB}}\Big|_{\mathscr{I}^-_+} = \sum_{j=0}^{\infty} (-2)^j \binom{n+j}{j} \overset{n+j+1,\ln}{F_{AB}}\Big|_{\mathscr{I}^+_-}. \tag{A.34}$$

# B   Maxwell equations at spatial infinity

In order to confirm the factor $(-1)^n$ in (60) which is absent in [29], we revisit the analysis presented there. The starting point is a foliation of the outer lightcone in Minkowski spacetime by three-dimensional de Sitter hyperboloids ($dS_3$),

$$ds^2 = d\rho^2 + \rho^2 h_{\alpha\beta}\, dx^\alpha\, dx^\beta, \tag{B.1}$$

with radial coordinate $\rho = \sqrt{X^\mu X_\mu}$. The components of the field strength are expanded at large $\rho$, i.e., in a neighborhood of spatial infinity $i^0$. Maxwell equations are found to impose hyperbolic second order differential equations on the resulting tensor fields, among which equation (5.10) in [29], namely

$$D^\alpha \overset{n}{F}_\alpha = 0, \qquad \left[D^2 + (n^2 - 2)\right] \overset{n}{F}_\alpha = 0. \tag{B.2}$$

Here $\overset{n}{F}_\alpha$ is a vector field on the unit dS$_3$ hyperboloid, with $D_\alpha$ the corresponding covariant derivative. It is denoted $\overset{n}{F}_{\rho\alpha}$ in [29] and appears in the asymptotic expansion near $i^0$ as mentioned above. Instead of the coordinates used in [29], we now cover the unit dS$_3$ hyperboloid with coordinates $x^\alpha = (\tau, x^A)$ and corresponding metric

$$h_{\alpha\beta}\, dx^\alpha\, dx^\beta = -d\tau^2 + \cosh^2 \tau\, \gamma_{AB}\, dx^A dx^B, \tag{B.3}$$

where $x^A$ are generic coordinates on the unit sphere with metric $\gamma_{AB}$ and covariant derivative $\nabla_A$. We then need to explicitly solve the dynamical equations (B.2), which we do following the methodology of [50]. The non-zero Christoffel symbols are given by

$$\Gamma^\tau_{AB} = \tanh \tau\, h_{AB}, \qquad \Gamma^A_{B\tau} = \tanh \tau\, \delta^A_B, \qquad \Gamma^C_{AB} = \Gamma^C_{AB}[\gamma], \tag{B.4}$$

such that

$$D^\alpha \overset{n}{F}_\alpha = \frac{1}{\sqrt{-h}} \partial_\alpha \left(\sqrt{-h}\, h^{\alpha\beta} \overset{n}{F}_\beta\right) = \cosh^{-2} \tau \left[-\partial_\tau(\cosh^2 \tau\, \overset{n}{F}_\tau) + \nabla^A \overset{n}{F}_A\right], \tag{B.5}$$

and

$$D^2 \overset{n}{F}_\tau = \left(-\partial_\tau^2 - 2\tanh \tau\, \partial_\tau + 2\tanh^2 \tau + \cosh^{-2} \tau\, \nabla^2\right) \overset{n}{F}_\tau - 2\tanh \tau \cosh^{-2} \tau\, \nabla^A \overset{n}{F}_A. \tag{B.6}$$

Hence, the equations (B.2) imply

$$\nabla^A \overset{n}{F}_A = \cosh^2 \tau\, (\partial_\tau + 2\tanh \tau)\, \overset{n}{F}_\tau, \tag{B.7}$$

and

$$\left[-\partial_\tau^2 - 4\tanh \tau\, \partial_\tau - 2\tanh^2 \tau + \cosh^{-2} \tau\, \nabla^2 + (n^2 - 2)\right] \overset{n}{F}_\tau = 0. \tag{B.8}$$

We now change the time variable $s = \tanh \tau \in (-1, 1)$ such that the latter becomes

$$\left[(1 - s^2)\partial_s^2 + 2s\partial_s - \nabla^2 + \frac{2s^2 + 2 - n^2}{1 - s^2}\right] \overset{n}{F}_\tau = 0. \tag{B.9}$$

To solve this equation, we decompose $\overset{n}{F}_\tau$ in spherical harmonics,

$$\overset{n}{F}_\tau = (1 - s^2) \sum_{lm} \overset{n}{F}_{lm}(s) Y_l^m(n^i), \tag{B.10}$$

such that the coefficients must satisfy the ordinary differential equation

$$\left[(1 - s^2)\partial_s^2 - 2s\partial_s + l(l+1) - \frac{n^2}{1 - s^2}\right] \overset{n}{F}_{lm}(s) = 0, \tag{B.11}$$

whose solutions are the Legendre functions $P_l^n(s)$ and $Q_l^n(s)$. In the limit $\tau \to \infty$ and for $l \geq n$, they have the asymptotic behavior

$$P_l^n(s) = O((1-s)^{n/2}) = O(e^{-n\tau}), \qquad Q_l^n(s) = O((1-s)^{-n/2}) = O(e^{n\tau}). \tag{B.12}$$

As suggested by the analysis in [29] we discard the growing solutions $Q_l^n(s)$. The parity properties of $P_l^n(s)$ and $Y_l^m(n^i)$ then implies the antipodal matching relation

$$\overset{n}{F}_\tau(-\tau, -n^i) = (-1)^n \overset{n}{F}_\tau(\tau, n^i). \tag{B.13}$$

To explicitly compare with [29], we need to use their time coordinate $\tilde{\tau} = \sinh\tau$. We thus find

$$\overset{n}{F}_{\tilde{\tau}} = \cosh^{-1}\tau \, \overset{n}{F}_\tau = O(\tilde{\tau}^{-n-3}), \tag{B.14}$$

whose behavior at large time is in agreement with the findings of [29]. Dropping tildes and following these authors, we introduce the late- and early-time expansions

$$\begin{aligned}
\overset{n}{F}_\tau &= \tau^{-n-3} \overset{n,+}{F}(n^i) + \dots && (\tau \to \infty), \\
\overset{n}{F}_\tau &= \tau^{-n-3} \overset{n,-}{F}(n^i) + \dots && (\tau \to -\infty),
\end{aligned} \tag{B.15}$$

such that the parity relation (B.13) now reads

$$\overset{n,-}{F}(-n^i) = -\overset{n,+}{F}(n^i). \tag{B.16}$$

As a last step, we turn to retarded and advanced coordinates

$$R = \rho\sqrt{1+\tau^2}, \qquad U = \rho\left(\tau - \sqrt{1+\tau^2}\right), \qquad V = \rho\left(\tau + \sqrt{1+\tau^2}\right), \tag{B.17}$$

such that the components $F_{UR}$ and $F_{VR}$ of interest are related to $F_{\rho\tau}$ by

$$F_{\rho\tau} = \frac{\rho}{\sqrt{1+\tau^2}} F_{RU} = \frac{\rho}{\sqrt{1+\tau^2}} F_{RV}. \tag{B.18}$$

In the limit $\tau \to \infty$ we have

$$U^{-1} = \rho^{-1}(-2\tau + \dots), \qquad U/R = -\frac{1}{2\tau^2} + \dots, \tag{B.19}$$

while for $\tau \to -\infty$ we have

$$V^{-1} = \rho^{-1}(-2\tau + \dots), \qquad V/R = \frac{1}{2\tau^2} + \dots \tag{B.20}$$

Plugging this in the expansion (5.17) in [29], namely

$$F_{RU} = \sum_{\ell=0}^{\infty}\sum_{k=\ell}^{\infty}(1/u)^{\ell+2}(u/r)^{k+2}\overset{k,\ell}{F}_{RU}(n^i), \qquad F_{RV} = \sum_{\ell=0}^{\infty}\sum_{k=\ell}^{\infty}(1/v)^{\ell+2}(v/r)^{k+2}\overset{k,\ell}{F}_{RV}(n^i), \tag{B.21}$$

one makes the identification

$$\overset{n,+}{F}(n^i) = \overset{n,n}{F}_{RU}(n^i), \qquad \overset{n,-}{F}(n^i) = (-1)^{n+1}\overset{n,n}{F}_{RV}(n^i). \tag{B.22}$$

Thus, the antipodal matching (B.16) amounts to

$$\overset{n,n}{F}_{RU}(-n^i)\Big|_{\mathscr{I}^+_-} = (-1)^n \overset{n,n}{F}_{RV}(n^i)\Big|_{\mathscr{I}^-_+}, \tag{B.23}$$

thereby confirming our result (60) in the case $k = n$.

# C   Maxwell equations at null infinity

Combining the results of the Appendix A, we find the following field strength components up to order $O(e^3)$:

$$F_{UR} = \frac{1}{R^2}\underbrace{\left(\overset{0,0}{F}_{UR} + O\left(e^3, U^{-1}\right)\right)}_{\overset{0}{F}_{UR}} + \frac{1}{R^3}\underbrace{\left(\overset{1,1}{F}_{UR} + U\overset{1,0}{F}_{UR} + \ln|U|\overset{1,\ln}{F}_{UR} + O(e^3, U^0)\right)}_{\overset{1}{F}_{UR}} + O(1/R^4), \quad \text{(C.1)}$$

$$F_{UA} = \underbrace{O(e^3, U^{-2})}_{\overset{0}{F}_{UA}} + \frac{1}{R}\underbrace{\left(\overset{1,1}{F}_{UA} + O(e^3, U^{-1})\right)}_{\overset{1}{F}_{UA}}$$

$$+ \frac{1}{R^2}\underbrace{\left(\overset{2,2}{F}_{UA} + U\overset{2,1}{F}_{UA} + \ln|U|\overset{2,\ln}{F}_{UA} + O(e^3, U^0)\right)}_{\overset{2}{F}_{UA}} + O(1/R^3), \quad \text{(C.2)}$$

$$F_{AB} = \underbrace{\left(\overset{0,0}{F}_{AB}\big|_{\mathscr{I}^+} + O(e^3, U^{-1})\right)}_{\overset{0}{F}_{AB}} + \frac{1}{R}\underbrace{\left(U\overset{1,0}{F}_{AB}\big|_{\mathscr{I}_-^+} + \overset{1,1}{F}_{AB}\big|_{\mathscr{I}_-^+} + \ln|U|\overset{1,\ln}{F}_{AB}\big|_{\mathscr{I}_-^+} + O(e^3, U^0)\right)}_{\overset{1}{F}_{AB}} + O(1/R^2),$$

$$\text{(C.3)}$$

$$F_{RA} = \frac{1}{R^2}\underbrace{\left(\overset{1,1}{F}_{RA}\big|_{\mathscr{I}_-^+} + U\overset{1,0}{F}_{RA}\big|_{\mathscr{I}_-^+} + \ln|U|\overset{1,\ln}{F}_{RA}\big|_{\mathscr{I}_-^+} + O(e^3, U^0)\right)}_{\overset{1}{F}_{RA}}$$

$$+ \frac{1}{R^3}\underbrace{\left(\overset{2,2}{F}_{RA}\big|_{\mathscr{I}_-^+} + U\overset{2,1}{F}_{RA}\big|_{\mathscr{I}_-^+} + U^2\overset{2,0}{F}_{RA}\big|_{\mathscr{I}_-^+} + U\ln|U|\overset{2,\ln}{F}_{RA}\big|_{\mathscr{I}_-^+} + O(e^3, U)\right)}_{\overset{2}{F}_{RA}} + O(1/R^4). \quad \text{(C.4)}$$

In retarded spherical coordinates, flat space is described by the metric

$$ds^2 = -dU^2 - 2dUdR + R^2 d\Omega_2^2, \quad \text{(C.5)}$$

with $d\Omega_2^2$ the standard metric on the two-sphere. In these coordinates, the Maxwell equations $\nabla^\alpha F_{\alpha\mu} = e^2 j_\mu$ become

$$-\partial_R\left(R^2 F_{UR}\right) + R^2 \partial_U F_{UR} - D^A F_{UA} = e^2 R^2 j_U, \quad \text{(C.6)}$$

$$-\partial_R(R^2 F_{UR}) + D^A F_{RA} = e^2 R^2 j_R, \quad \text{(C.7)}$$

$$-R^2 \partial_U F_{RA} + R^2 \partial_R\left(F_{RA} - F_{UA}\right) - D^B F_{AB} = e^2 R^2 j_A. \quad \text{(C.8)}$$

These equations are supplemented by the Bianchi identities $\partial_{[\mu} F_{\nu\rho]} = 0$. They match in stereographic coordinates with Eq. (2.3) of [24]. They match with [29] after inverting the convention for the current $j_\mu \mapsto -j_\mu$. We also have current conservation

$$\partial_R(R^2 j_R + R^2 j_U) + R^2 \partial_U j_R + D^A j_A = 0. \quad \text{(C.9)}$$

Assuming the falloffs

$$j_U = R^{-2}\overset{0}{j}_U + O(R^{-3}), \qquad j_R = R^{-4}\overset{0}{j}_R + O(R^{-5}), \qquad j_A = R^{-2}\overset{0}{j}_A + O(R^{-3}), \quad \text{(C.10)}$$

the leading order of the Maxwell equations is given by

$$-\partial_U \overset{0}{F}_{UR} + D^A \overset{0}{F}_{UA} = -e^2 \overset{0}{j}_U\,, \tag{C.11}$$

$$-\overset{1}{F}_{UR} + D^A \overset{1}{F}_{RA} = -e^2 \overset{0}{j}_R\,, \tag{C.12}$$

$$\partial_U \overset{1}{F}_{RA} - \overset{1}{F}_{UA} + D^B \overset{0}{F}_{AB} = -e^2 \overset{0}{j}_A\,. \tag{C.13}$$

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
