# Peer review of "Electromagnetic multipole expansions and the logarithmic soft photon theorem"

_SciPost Physics, doi:SciPost Phys. Core 8, 066 (2025)_

## Round 2 · Referee Report · Anonymous (Referee 1) · 2025-5-20

Strengths

  1. Nicely written manuscript
  2. Summarizes the results of ref [29,30,31,33,34]

Weaknesses

  1. The manuscript does not contain any new results.
  2. The main results of the manuscript have already been addressed in ref [37] and proved in arXiv: 2007.03627.

Report

The authors claim that they are the first to establish the conservation laws at spatial infinity related to the logarithmic soft theorem, distinct from the proposal in Ref. [29]. However, it should be noted that the main result of the paper—the conservation law presented in Eq. (4.1)—has already been used in Ref. [37] to derive logarithmic soft photon theorem and was also independently proved in Section 3 of arXiv:2007.03627.

Given that the central result of this work has been previously established in the literature, the manuscript does not offer sufficient novelty to merit publication in this journal.

Recommendation

Reject

---

## Round 2 · Referee Report · Anonymous (Referee 2) · 2025-8-5

Strengths

  1. Approach is quite general and so of relatively broad applicability
  2. Makes contact with different approaches and results in the literature
  3. Paper is well written and clearly lays out the details of the computation

Weaknesses

  1. The approach is quite technical and the explicit results overlap with those already known in the literature such that the significance of the approach is not immediately apparent.

Report

Understanding the behaviour of classical fields in asymptotically flat space-times, and their relation to conservation laws, has become an important topic in recent years. In this work, the authors use a multipolar decomposition to derive antipodal matching conditions for the electromagnetic field across spatial infinity. This is done at leading and subleading order in the electric charge, allowing them to derive the classical logarithmic soft-photon theorem. The paper builds on, and overlaps with, the results of references [30], [38], and [20]; the relationship to this earlier work is clarified in the updated version 3 of the arXiv submission.

While some of the explicit results were previously known, the perspective and approach here are sufficiently distinct — particularly through the connection to the general multipole expansion of Damour and Iyer — that the work provides a useful contribution to the topic and is deserving of publication. One concrete, albeit straightforward, new result is the extension of the matching conditions for logarithmic terms to include terms with additional polynomial powers of the expansion coordinates. More broadly, this represents a promising approach to the problem of asymptotic boundary conditions and conserved charges. The paper is well written and clearly presents what is a quite technical computation.

Recommendation

Publish (meets expectations and criteria for this Journal)

---

## Editorial Decision

published